# Caesarean section trends in Catalonia between 2013 and 2017 based on the Robson classification system: A cross-sectional study

**Garazi Carrillo-Aguirre**[1,2], **Albert Dalmau-Bueno**[2], **Carlos Campillo-Artero**[3], **Anna García-Altés**[2,4,5]*

**1** Master's Degree Public Health, Pompeu Fabra University (UPF), Barcelona, Spain, **2** Department of Health, Health Evaluation and Quality Agency of Catalonia (AQuAS), Government of Catalonia, Barcelona, Catalonia, Spain, **3** Center for Research in Health and Economics, Pompeu Fabra University, IB-Salut, Barcelona, Spain, **4** Biomedical Research Networking Center for Epidemiology and Public Health (CIBERESP), Madrid, Spain, **5** Sant Pau Biomedical Research Institute (IIB Sant Pau), Barcelona, Spain

* agarciaaltes@gencat.cat

## Abstract

### Introduction

In Catalonia caesarean rates have always been analysed as a single percentage. The objective is to estimate caesarean section rates using the Robson classification in publicly funded hospitals in Catalonia between 2013 and 2017, considering sociodemographic, institutional and obstetric characteristics.

### Materials and methods

Cross-sectional population-based study in Catalonia including all women delivering within publicly funded hospitals between 2013–2017 (n = 210 020). The modified Robson classification distribution was estimated, the caesarean rate and the overall contribution, analysed for each year, and by confounders, through logistic regression models.

### Results

CS rates decreased steadily between 2013 and 2017 in Catalonia within publicly funded hospitals from 24.3% to 22.8% (cOR 0.92, 95% CI; 0.89 to 0.95). Once adjusted for changes in sociodemographic, institutional and obstetric characteristics the observed decline was even more pronounced (aOR 0.87, 95% CI; 0.84 to 0.90). Within the different groups of Robson once adjusted for confounders, groups 1+2 (aOR 0.88, 95% CI; 0.83 to 0.93), 3+4 (aOR 0.83, 95% CI; 0.78 to 0.89) and 10 (aOR 0.78, 95% CI; 0.68 to 0.90) presented a reduction in caesarean section rates, whereas group 5 showed no significant decrease (aOR 0.95, 95% CI; 0.87 to 1.03%).

### Conclusions

The decrease in caesarean section rates in Catalonia is more pronounced when adjusted for known confounders, suggesting retrospective overutilization of caesarean section and

**Data Availability Statement:** Data cannot be shared publicly because of confidentiality. Data are available from the Government of Catalonia

Institutional Data Access / Ethics Committee for researchers who meet the criteria for access to confidential data. The data underlying the results presented in the study are available from the Government of Catalonia Registries. Plos One can access aggregated data at hospital level and values used to build graphs under request. You can check up-to-date regulations at http://aquas.gencat.cat/ca/ambits/analitica-dades/padris/ and address any inquiries to AQuAS director, Dr. Cesar Velasco (cesarvelasco@gencat.cat).

**Funding:** The authors received no specific funding for this work.

**Competing interests:** The authors have declared that no competing interests exist.

percentages of (in)adequacy in the past. In any case, it remains above the recommended by experts. Further efforts should be made to achieve optimum rates, including improvement on obstetric data collection

## Introduction

A Caesarean Section (CS) is a surgical procedure that, when performed for medical reasons, could save the life of a woman and her baby. However, it carries risks for both of them and therefore should only be considered when necessary [1,2].

The increase on the global rates of CS remains a continuing public health concern [3,4]. The World Health Organization (WHO) stated that priority should remain the provision of CS to all women in need, rather than the achievement of an ideal level [5]. In places where CS is universally available though, optimal rates should be expected (i.e. Catalonia). Not excepted of controversy, recent studies suggested rates between 10–20% of all births [6–9].

There are complex reasons behind the significant increase in CS [10], believed, in principle, to correlate with higher risk profiles of pregnant women and their babies [11]. However, Betrán et al. suggested that it responds to a multifactorial phenomenon including healthcare organisations as well as financial aspects among others [12].

During the years, the comparison of national rates has mask inequalities on access and practice. The WHO (2015) proposed the Robson classification as a global standard system for assessing, monitoring and comparing CS rates [5]. This classifies women into ten different groups, all mutually exclusive, and totally inclusive, based on obstetric characteristics [13].

In Catalonia, the CS rate have concerned health authorities for very long time. Monitoring of CS rate started in 1990 and different published National Health Plans have included the objective of CS reduction [14–18]. However, despite having also implemented several protocols and guidelines in regards to care during delivery and the publication of official reports comparing CS rates between hospitals [19–24], their impact is unknown due to a lack of exhaustive evaluation assessments. Rates have ranged between 22% and 32%, with considerable differences between the public and the private sectors, i.e. 22.3% to 35.9% for the year 2017 [25]. Maternal and neonatal mortality ratio has remained very low (MMR: 3.1, 2010–2014 [26] and NMR: 1.67, 2014–2017 [27]) and public hospitals do not provide CS under maternal request [28]. In this context, this analysis provides an exciting opportunity for a deeper understanding of the CS rate fluctuation.

Motivated by the opportunity that the Robson classification provides, this current study analyses CS trends between 2013 and 2017, in order to firstly, identify the groups of women with the highest contribution to CS rates, and secondly, observe changes in the total CS rate by adjusting for sociodemographic, institutional and obstetric characteristics. The results hope to provide valuable information for the establishment of public health benchmarks for maternal and neonatal health programs, as well as the design of policies and guidelines.

## Material and methods

### Study design and participants

A retrospective cross-sectional trend study was conducted in the Spanish region of Catalonia. The population included women delivering between 1st January 2013 and 31st December 2017. From the total 330,851 (100%) deliveries occurred after 22 weeks gestation, only those that occurred at the 44 publicly funded hospitals offering maternity care in Catalonia were

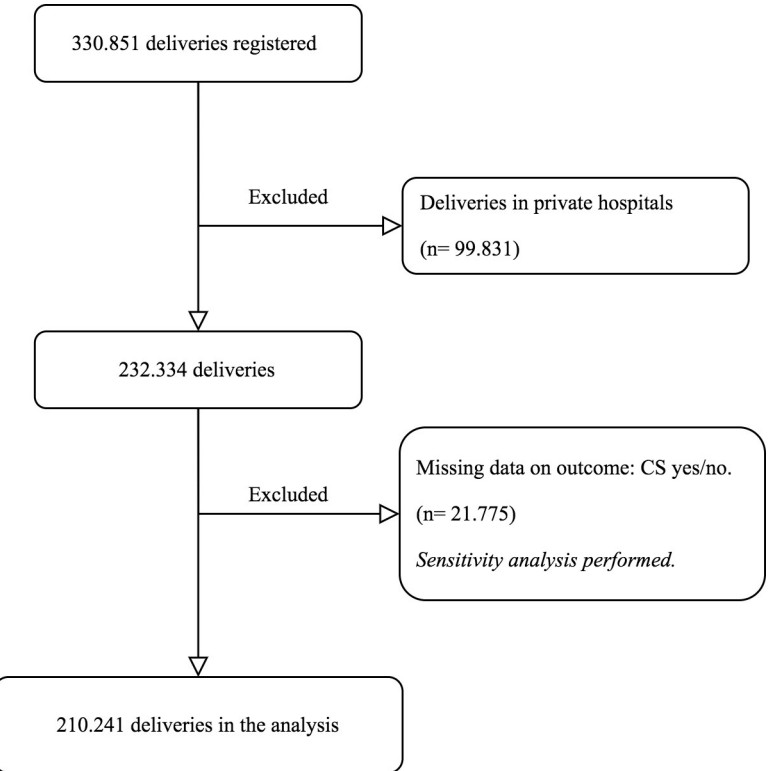

**Fig 1. Flowchart of the population included in the study.**

included. As a result, 231,020 (69.83%) deliveries met the eligibility criteria for this research. However, information regarding type of delivery, whether it was vaginal or caesarean section, was considered a *sine qua non* variable for the study. Thus, missing data in 9.1% of the women regarding outcome (vaginal/CS), reduced the potential sample size to 210,241 (Fig 1).

## Details of ethics approval

A governmental structure called PADRIS exists within the Catalan healthcare administration. PADRIS is an analytical data programme for research and innovation in health. The purpose of the structure is to make data available to scientific communities to promote research, innovation, and evaluation in health, with the aim of reusing and exchanging the data generated by the health system in accordance with the legal and regulatory framework. The programme PADRIS ensures that all data made available to researchers is fully anonymised. As workers within the Catalan healthcare administration, we are provided data by PADRIS, and therefore are never given access to identifiable information. Thus, the study did not involve any data collection, requiring neither human participants nor patient consent. For that reason, and due to the use of existing anonymised data for research, the study was exempt from institutional review committee approval. It is the standard way of proceeding in the healthcare administration to systematically check the quality of the healthcare providers in our context.

## Data sources

Three data sources were employed: (1) Minimum Basic Data Set (*Conjunt Mínim Bàsic de Dades*, CMBD); (2) the Central Registry of Insured Persons (*Registro Central de personas*

*Aseguradas*, RCA); and (3) the Clinic Station for Primary Care (*Estació Clínica d' Atenció Primària*, E-CAP).

Individuals on the registries were given a personal anonymised code that allowed linkage between databases. The different databases were then merged using this anonymised code.

1. CMBD is an administrative registry subject to Ministerial regulation containing exhaustive data regarding hospital discharges. All hospitals are required to provide information regarding hospital activity and diagnosis (ICD-9-CM at the time of the current study) [29]. This registry allowed access to information regarding type of delivery and the majority of variables required for the Robson classification.

2. RCA collects personal data of all those insured by the CatSalut (Catalan National Health System), including income level and employment status [30], enabling access to socioeconomic status.

3. E-CAP is a computerized system used within the primary healthcare facilities. This enabled access to details of Body Mass Index (BMI) and parity [31].

### Definition of variables

The primary outcome for this study was delivery by CS. The following variables were required to create the variable Robson classification: parity, gestational age, foetal presentation, previous CS, number of foetuses, and onset of labour [32]. The variable foetal presentation was not recorded within the databases, so it had to be created ad hoc, thus considering cephalic presentation in the absence of breech or transverse lie. A modified classification was constructed due to onset of labour not being collected in any of the databases.

An adapted Robson classification was created formed of eight mutually exclusive categories. This combined groups 1 and 2, and 3 and 4, with the rest remaining unchanged: nulliparous, singleton cephalic pregnancy, $\geq 37$ weeks' gestation (groups 1+2); multiparous without a previous uterine scar, with singleton, cephalic pregnancy, $\geq 37$ weeks' gestation (groups 3+4); previous CS, singleton, cephalic, $\geq 37$ weeks' gestation (Group 5); all nulliparous with a single breech (Group 6); all multiparous with a single breech, including previous CS (Group 7); all multiple pregnancies, including previous CS (Group 8); all women with a single pregnancy in transverse or oblique lie (including those with previous CS (Group 9); all singleton, cephalic, $< 37$ weeks' gestation pregnancies, including previous CS (Group 10).

The following variables were also included: age at the time of delivery, nationality, socioeconomic status, hospital level, BMI and maternal complications. For the variable nationality The World Bank classification was selected [33]. The variable socioeconomic status is a set variable within the database that arises through pharmaceutical co-pay estimations. There are four co-payment groups: 1) those exempt from co-payment, (disadvantaged population; individuals receiving some form of universal pension scheme, those who no longer receive unemployment allowance, or those who no longer receive unemployment benefit and do not qualify for unemployment allowance); 2) those with annual earnings of less than 18,000€; 3) those with annual earnings between 18,000€ and 100,000€; and 4) those with annual earnings over 100,000€. This is predetermined by the system and does not allow further disaggregation. The hospital level follows an order of complexity, based on the care that the pregnant woman and her baby might require. Thus, even though all hospitals will be caring for women with low risk pregnancies, the more complex the pregnancy becomes the higher the level of the hospital in which she will be cared for. Level IA being the less medicalised hospitals, and IIIB the most, offering medical attention from other specialties if required [34]. The variable maternal complication was created ad hoc and included any of the following diagnoses that had to be recorded prior initiating the labour process within this pregnancy: uterine rupture, placenta praevia, pre-

eclampsia, gestational diabetes, diabetes mellitus, heart disease, hypertension, hepatic disease, viral disease, anaemia, renal disease and epilepsy. A code recorded at CMBD that includes different diagnosis called "disease that complicates the pregnancy" was also included (For further information S1 Table).

## Data analysis

**Missing data.** Missing data relating to the different variables ranged between 0.2 and 32%, with CS showing an overall level of 6%. Maternal age accounted for 0.2%; nationality 1.2%; socioeconomic status 0.2%; healthcare facility 1.6%; and BMI 32%. To minimize loss of statistical power, BMI was not included in the final model and a separate sensibility analysis was undertaken. An analysis of the missing data concerning type of delivery was also undertaken to determine any bias related to the response rates.

**Statistical analysis.** Characteristics of the women included in the study were reported for each year, along with the proportion of women delivered by CS. The following were analysed for each specific year and modified Robson classification: the relative size of the obstetric population (% = n of women in the group/total N women delivered x 100), total CS rate (% = n of CS in the group/total N of women in the group x 100) and the absolute contribution to the total CS rate (% = n of CS in the group/total N of women delivered). The crude change between 2013 and 2017 was calculated for each of the above.

Two logistic regression models were used for the analysis between the principal variable (CS) and the remainder of the variables. Crude and adjusted odds ratios were estimated with their correspondent 95% confidence intervals (CI) and p-values. Following the same analytical strategy, maintaining the CS as dependent variable, through ten regression logistic models, crude and adjusted odds ratios were also estimated for each year and for each Robson group for each year.

Finally, the performance of the logistic models applied to estimate time trends were assessed. On the one hand, the area under the receiver operating curve (ROC) was estimated to determine the discriminatory capacity and on the other hand, the calibration belt p-value for internal validation. This analysis pretended to establish a reliable baseline. The aim of the study was to analyse CS trends and the intention was not to build a predictive model for CS, although the possibility was explored. Stata software version 14 was used to perform the analysis.

## Results

A significant decrease was identified in CS rates between 2013 and 2017, from 24.30% to 22.80% (see Table 1). The majority of the women on the population sample were Spanish (62.43%), with an income of less than 18.000€ (76.15%). Their average age was 30 (SD 5.6), with most having singleton, cephalic births at term. Some characteristics changed slightly over the years, with the proportion of births to women aged thirty-five or over increasing by 5.23% and an overall significant increase of women with a BMI above 30. In addition, the decrease in the proportion of nulliparous women and increase in multiparous women with previous CS were statistically significant. There was also an increase in the number of women presenting a diagnosis potentially complicating the pregnancy.

### Determinants of caesarean sections

During the study period, 50.009 women were delivered by CS. Rates increased in relation to maternal age, women under twenty with a rate of 14.59%, while those over forty 37.18% (S2 Table). Adjusting for sociodemographic, institutional and obstetric factors, the age differences

**Table 1. Socio-demographic and obstetric characteristics in the study population for each year, 2013–2017.**

| | Year 2013 n = 43145 | | | Year 2014 n = 42397 | | | Year 2015 n = 41825 | | | Year 2016 n = 41275 | | | Year 2017 n = 41599 | | |
|---|---|---|---|---|---|---|---|---|---|---|---|---|---|---|---|
| | n | % | 95% CI | n | % | 95% CI | n | % | 95% CI | n | % | 95% CI | n | % | 95% CI |
| **Type of delivery** | | | | | | | | | | | | | | | |
| Vaginal | 32659 | 75.7 | 75.29–76.10 | 32134 | 75.79 | 75.38–76.20 | 31802 | 76.04 | 75.62–76.44 | 31526 | 76.37 | 75.96–76.78 | 32114 | 77.2 | 76.79–77.60 |
| CS | 10486 | 24.3 | 23.90–24.71 | 10263 | 24.21 | 23.80–24.62 | 10023 | 23.96 | 23.56–24.38 | 9752 | 23.63 | 23.22–24.04 | 9485 | 22.8 | 22.40–23.21 |
| **Parity** | | | | | | | | | | | | | | | |
| Nuliparous | 16486 | 38.21 | 37.75–38.67 | 15647 | 36.91 | 36.45–37.37 | 14790 | 35.36 | 34.90–35.82 | 14097 | 34.15 | 33.69–34.61 | 14274 | 34.31 | 33.86–34.77 |
| Multiparous | 26659 | 61.79 | 61.33–62.25 | 26750 | 63.09 | 62.63–63.55 | 27035 | 64.64 | 64.18–65.10 | 27181 | 65.85 | 65.39–66.31 | 27325 | 65.69 | 65.23–66.14 |
| **Previous CS** | | | | | | | | | | | | | | | |
| No | 39054 | 90.52 | 90.24–90.79 | 38266 | 90.26 | 89.97–90.54 | 37496 | 89.65 | 89.35–89.94 | 36889 | 89.37 | 89.07–89.66 | 37116 | 89.22 | 88.92–89.52 |
| Yes | 4091 | 9.48 | 9.21–9.76 | 4131 | 9.74 | 9.46–10.03 | 4329 | 10.35 | 10.06–10.65 | 4389 | 10.63 | 10.34–10.93 | 4483 | 10.78 | 10.48–11.08 |
| **Maternal age years** | | | | | | | | | | | | | | | |
| <20 | 1319 | 3.06 | 2.90–3.22 | 1218 | 2.87 | 2.72–3.04 | 1192 | 2.85 | 2.69–3.01 | 1159 | 2.81 | 2.65–2.97 | 1159 | 2.79 | 2.63–2.95 |
| 20–24.9 | 5040 | 11.68 | 11.38–11.99 | 4673 | 11.02 | 10.73–11.32 | 4498 | 10.75 | 10.46–11.06 | 4473 | 10.84 | 10.54–11.14 | 4393 | 10.56 | 10.27–10.86 |
| 25–29.9 | 10590 | 24.55 | 24.14–24.95 | 10370 | 24.46 | 24.05–24.87 | 9940 | 23.77 | 23.36–24.18 | 9600 | 23.26 | 22.85–23.67 | 9470 | 22.76 | 22.36–23.17 |
| 30–34.9 | 15028 | 34.83 | 34.38–35.28 | 14618 | 34.48 | 34.03–34.93 | 14008 | 33.49 | 33.04–33.95 | 13695 | 33.18 | 32.73–33.64 | 13632 | 32.77 | 32.32–33.22 |
| 35–39.9 | 9205 | 21.34 | 20.95–21.72 | 9503 | 22.41 | 22.02–22.81 | 9960 | 23.81 | 23.41–24.22 | 9895 | 23.97 | 23.56–24.39 | 10269 | 24.69 | 24.27–25.10 |
| ≥40 | 1963 | 4.55 | 4.36–4.75 | 2015 | 4.75 | 4.55–4.96 | 2227 | 5.32 | 5.11–5.54 | 2453 | 5.94 | 5.72–6.18 | 2676 | 6.43 | 6.20–6.67 |
| **Nationality** | | | | | | | | | | | | | | | |
| Spain | 27116 | 62.85 | 62.39–63.30 | 26428 | 62.33 | 61.87–62.80 | 26314 | 62.91 | 62.45–63.38 | 25503 | 61.78 | 61.31–62.25 | 25886 | 62.23 | 61.76–62.69 |
| Rest of Europe and Central Asia | 2591 | 6.01 | 5.78–6.23 | 2643 | 6.23 | 6.01–6.47 | 2712 | 6.48 | 6.25–6.72 | 2707 | 6.56 | 6.32–6.80 | 2835 | 6.82 | 6.57–7.06 |
| Middle East and North Africa | 5830 | 13.51 | 13.19–13.84 | 5442 | 12.84 | 12.52–13.16 | 5430 | 12.98 | 12.66–13.31 | 5559 | 13.47 | 13.14–13.80 | 5634 | 13.54 | 13.22–13.88 |
| Latin América and Caribbean | 4368 | 10.12 | 9.84–10.41 | 4555 | 10.74 | 10.45–11.04 | 4246 | 10.15 | 9.86–10.45 | 4302 | 10.42 | 10.13–10.72 | 3889 | 9.35 | 9.07–9.63 |
| East Asia and Pacific | 992 | 2.3 | 2.16–2.45 | 1007 | 2.38 | 2.23–2.52 | 802 | 1.92 | 1.79–2.05 | 841 | 2.04 | 1.90–2.18 | 790 | 1.9 | 1.77–2.03 |
| Subsaharian Africa | 1138 | 2.64 | 2.49–2.79 | 1130 | 2.67 | 2.51–2.82 | 1163 | 2.78 | 2.63–2.94 | 1118 | 2.71 | 2.55–2.87 | 1075 | 2.58 | 2.43–2.74 |
| South Asia | 1094 | 2.54 | 2.39–2.69 | 1152 | 2.72 | 2.56–2.88 | 1133 | 2.71 | 2.56–2.87 | 1216 | 2.95 | 2.78–3.11 | 1463 | 3.52 | 3.34–3.70 |
| North America | 16 | 0.04 | 0.02–0.06 | 40 | 0.09 | 0.07–0.13 | 25 | 0.06 | 0.04–0.09 | 32 | 0.08 | 0.05–0.11 | 27 | 0.06 | 0.04–0.09 |
| **Socioeconomic status** | | | | | | | | | | | | | | | |
| Disadvantaged population | 2516 | 5.83 | 5.61–6.06 | 2123 | 5.01 | 4.80–5.22 | 2200 | 5.26 | 5.05–5.48 | 2349 | 5.69 | 5.47–5.92 | 2270 | 5.46 | 5.24–5.68 |
| Income <18.000 | 32238 | 74.72 | 74.31–75.13 | 32517 | 76.7 | 76.29–77.10 | 32393 | 77.45 | 77.05–77.85 | 31703 | 76.8 | 76.39–77.21 | 31246 | 75.11 | 74.69–75.53 |
| Income 18.000–100.000 | 8369 | 19.4 | 19.03–19.77 | 7740 | 18.26 | 17.89–18.63 | 7208 | 17.23 | 16.87–17.60 | 7202 | 17.45 | 17.08–17.82 | 8057 | 19.37 | 18.99–19.75 |
| Income >100.000 | 22 | 0.05 | 0.03–0.08 | 17 | 0.04 | 0.02–0.06 | 24 | 0.06 | 0.04–0.09 | 24 | 0.06 | 0.04–0.09 | 26 | 0.06 | 0.04–0.09 |
| **Healthcare facility** | | | | | | | | | | | | | | | |
| Level IA | 13375 | 31 | 30.56–31.44 | 12757 | 30.09 | 29.65–30.53 | 12660 | 30.27 | 29.83–30.71 | 12140 | 29.41 | 28.97–29.85 | 11813 | 28.4 | 27.96–28.83 |

*(Continued)*

**Table 1.** (Continued)

| | Year 2013 n = 43145 | | | Year 2014 n = 42397 | | | Year 2015 n = 41825 | | | Year 2016 n = 41275 | | | Year 2017 n = 41599 | | |
|---|---|---|---|---|---|---|---|---|---|---|---|---|---|---|---|
| | n | % | 95% CI | n | % | 95% CI | n | % | 95% CI | n | % | 95% CI | n | % | 95% CI |
| Level IIA | 7070 | 16.39 | 16.04–16.74 | 7032 | 16.59 | 16.23–16.94 | 6694 | 16 | 15.65–16.36 | 6356 | 15.4 | 15.05–15.75 | 6406 | 15.4 | 15.05–15.75 |
| Level IIB | 6666 | 15.45 | 15.11–15.79 | 6602 | 15.57 | 15.23–15.92 | 6557 | 15.68 | 15.33–16.03 | 6591 | 15.97 | 15.62–16.32 | 7107 | 17.08 | 16.72–17.45 |
| Level IIIA | 10566 | 24.49 | 24.08–24.90 | 10526 | 24.83 | 24.42–25.24 | 10547 | 25.22 | 24.80–25.64 | 10577 | 25.62 | 25.20–26.05 | 10442 | 25.1 | 24.69–25.52 |
| Level IIIB | 5468 | 12.67 | 12.36–12.99 | 5480 | 12.93 | 12.61–13.25 | 5367 | 12.83 | 12.51–13.16 | 5614 | 13.6 | 13.27–13.93 | 5831 | 14.02 | 13.68–14.35 |
| **Fetal presentation** | | | | | | | | | | | | | | | |
| Cephalic | 41833 | 96.96 | 96.79–97.12 | 41123 | 97 | 96.83–97.16 | 40503 | 96.84 | 96.67–97.00 | 40040 | 97 | 96.83–97.16 | 40324 | 96.94 | 96.76–97.10 |
| Breech | 1122 | 2.6 | 2.45–2.76 | 1067 | 2.52 | 2.37–2.67 | 1154 | 2.76 | 2.60–2.92 | 1061 | 2.57 | 2.42–2.73 | 1090 | 2.62 | 2.47–2.78 |
| Oblique/Transverse | 190 | 0.44 | 0.38–0.51 | 207 | 0.49 | 0.42–0.56 | 168 | 0.4 | 0.34–0.47 | 177 | 0.43 | 0.37–0.50 | 185 | 0.44 | 0.38–0.51 |
| **Gestational age** | | | | | | | | | | | | | | | |
| ≥ 37 weeks term | 40492 | 93.85 | 93.62–94.08 | 39822 | 93.93 | 93.69–94.15 | 39383 | 94.16 | 93.93–94.38 | 38811 | 94.02 | 93.79–94.25 | 39194 | 94.22 | 93.99–94.44 |
| <37 weeks preterm | 2653 | 6.15 | 5.92–6.38 | 2575 | 6.07 | 5.85–6.31 | 2442 | 5.84 | 5.62–6.07 | 2467 | 5.98 | 5.75–6.21 | 2405 | 5.78 | 5.56–6.01 |
| **Pregnancy complication** | | | | | | | | | | | | | | | |
| No | 28616 | 66.33 | 65.88–66.77 | 27274 | 64.33 | 63.87–64.79 | 27054 | 64.68 | 64.22–65.14 | 26254 | 63.6 | 63.14–64.07 | 26266 | 63.14 | 62.68–63.60 |
| Yes | 14529 | 33.67 | 33.23–34.12 | 15123 | 35.67 | 35.21–36.13 | 14771 | 35.32 | 34.86–35.78 | 15024 | 36.4 | 35.93–36.86 | 15333 | 36.86 | 36.40–37.32 |
| **Number of neonates** | | | | | | | | | | | | | | | |
| Singleton | 42360 | 98.18 | 98.05–98.30 | 41589 | 98.09 | 97.96–98.22 | 41047 | 98.14 | 98.01–98.27 | 40426 | 97.94 | 97.79–98.07 | 40781 | 98.03 | 97.90–98.16 |
| Multiple | 785 | 1.82 | 1.70–1.95 | 808 | 1.91 | 1.78–2.04 | 778 | 1.86 | 1.73–1.99 | 852 | 2.06 | 1.93–2.21 | 818 | 1.97 | 1.84–2.10 |
| **Robson classification** | | | | | | | | | | | | | | | |
| Group 1+2 | 14505 | 33.62 | 33.17–34.07 | 13761 | 32.46 | 32.01–32.91 | 12940 | 30.94 | 30.50–31.38 | 12379 | 29.99 | 29.55–30.43 | 12542 | 30.15 | 29.71–30.59 |
| Group 3+4 | 20335 | 47.13 | 46.66–47.60 | 20425 | 48.18 | 47.70–48.65 | 20616 | 49.29 | 48.81–49.77 | 20678 | 50.09 | 49.61–50.58 | 20723 | 49.82 | 49.33–50.30 |
| Group 5 | 4210 | 9.76 | 9.48–10.04 | 4203 | 9.91 | 9.63–10.20 | 4304 | 10.29 | 10.00–10.59 | 4316 | 10.46 | 10.16–10.76 | 4455 | 10.71 | 10.41–11.01 |
| Group 6 | 532 | 1.23 | 1.13–1.34 | 499 | 1.18 | 1.08–1.28 | 526 | 1.26 | 1.15–1.37 | 431 | 1.04 | 0.95–1.15 | 450 | 1.08 | 0.98–1.19 |
| Group 7 | 487 | 1.13 | 1.03–1.23 | 469 | 1.11 | 1.01–1.21 | 525 | 1.26 | 1.15–1.37 | 507 | 1.23 | 1.12–1.34 | 524 | 1.26 | 1.15–1.37 |
| Group 8 | 785 | 1.82 | 1.70–1.95 | 808 | 1.91 | 1.78–2.04 | 778 | 1.86 | 1.73–1.99 | 852 | 2.06 | 1.93–2.21 | 818 | 1.97 | 1.84–2.10 |
| Group 9 | 159 | 0.37 | 0.31–0.43 | 178 | 0.42 | 0.36–0.49 | 145 | 0.35 | 0.29–0.41 | 140 | 0.34 | 0.29–0.40 | 163 | 0.39 | 0.33–0.46 |
| Group 10 | 2132 | 4.94 | 4.74–5.15 | 2054 | 4.84 | 4.64–5.05 | 1991 | 4.76 | 4.56–4.97 | 1975 | 4.78 | 4.58–4.99 | 1924 | 4.63 | 4.43–4.83 |
| | n = 29804 | | | n = 29602 | | | n = 29257 | | | n = 29272 | | | n = 29338 | | |
| **Body Mass Index BMI** | | | | | | | | | | | | | | | |
| <18.5 | 533 | 1.79 | 1.64–1.95 | 590 | 1.99 | 1.84–2.16 | 586 | 2 | 1.85–2.17 | 580 | 1.98 | 1.82–2.15 | 605 | 2.06 | 1.90–2.23 |
| 18.5–24.9 | 13496 | 45.28 | 44.72–45.85 | 13504 | 45.62 | 45.05–46.19 | 13292 | 45.43 | 44.86–46.00 | 13085 | 44.7 | 44.13–45.27 | 12732 | 43.4 | 42.83–43.97 |
| 25–29.9 | 10100 | 33.89 | 33.35–34.43 | 9894 | 33.42 | 32.89–33.96 | 9753 | 33.34 | 32.80–33.88 | 9669 | 33.03 | 32.49–33.57 | 9847 | 33.56 | 33.02–34.11 |
| 30–34.9 | 4047 | 13.58 | 13.19–13.97 | 4018 | 13.57 | 13.19–13.97 | 4034 | 13.79 | 13.39–14.19 | 4252 | 14.53 | 14.12–14.93 | 4364 | 14.87 | 14.47–15.29 |
| 35–39.9 | 1205 | 4.04 | 3.82–4.27 | 1178 | 3.98 | 3.76–4.21 | 1192 | 4.07 | 3.85–4.31 | 1240 | 4.24 | 4.01–4.47 | 1345 | 4.58 | 4.35–4.83 |
| ≥40 | 423 | 1.42 | 1.29–1.56 | 418 | 1.41 | 1.28–1.55 | 400 | 1.37 | 1.24–1.51 | 446 | 1.52 | 1.39–1.67 | 445 | 1.52 | 1.38–1.66 |

remain similar. In particular, women between 35 and 40 presented a 37% higher chance of CS, and women over 40, twice the likelihood of having a CS than women aged between 25 and 30 (Table 2).

In comparison to Spanish women (24.45%), higher rates of CS were found in women from sub-Saharan Africa (27.19%), Latin America and the Caribbean (27.27%) and South Asia (27.89%) (S2 Table). These groups remained similar after adjusting for confounders (Table 2).

The variable of socioeconomic level revealed that women with an income of over 18.000€ showed less probability of CS. Differences were also noted relating to the complexity level of the hospital, with Level IIIB undertaking the highest number and Level IA the least (Level IIIB compared to Level IA, crude OR 1.37, 95% CI 1.33–1.42). However, these differences can be explained when adjusting for confounders (Level IIIB compared to Level IA OR 1.07, 95% CI 1.03–1.11). Moreover, as expected, women with any diagnosis potentially complicating the pregnancy had a higher probability of CS, even when adjusting for other confounders (OR 1.44, 95% CI 1.41–1.47) (Table 2).

## Trends over time in caesarean section rates among the modified Robson groups

The overall CS rate steadily declined between 2013 and 2017. The following figures show each of the obstetric groups according to the modified Robson classification during the five years of the study period: Fig 2, the CS rate; Fig 3, the relative size of the obstetric population; and Fig 4, the absolute contribution to the overall CS rate.

Within the population sample, the majority of women carried single babies at term in cephalic presentation and belonged to groups 1+2 and 3+4 (31.45% and 48.88%, respectively), followed by multiparous women with a uterine scar (10.22%) and women with premature babies (4.79%).

The CS rate was lowest for groups 1+2 and 3+4 and highest (i.e. almost 100%) for groups 6, 7 and 9. Group 5 had an average rate of 54.73%; Group 8 63.33% and Group 10 35.34% (see Fig 2). Most groups demonstrated a subtle downwards tendency in CS rates observed in the graph.

The major contributors to the absolute CS rates were groups 1+2, 3+4 and 5. The considerable reduction in group 1+2 was due to the reduced group size and CS rates. By contrast, group 5 showed an increased overall contribution due to the increasement on the size.

It has been established that high-risk pregnancies are a risk factor for CS. The initial analyses revealed that the probability of CS in 2017 was, in comparison to 2013, reduced by 8% (OR 0.92, 95% CI 0.89–0.95) (Table 3). However, the profile of women described in Table 1 revealed a significant increase in known risk factors for CS (i.e. age or pregnancy complications). Thus, the analysis adjusted by confounders showed that there was even less probability in comparison to 2013 i.e. 13% (OR 0.87, 95% CI 0.84–0.90). The significant reduction in CS rates was already observed in 2015 (OR 0.96, CI 95% 0.92–0.99) and maintained throughout 2016 (OR 0.93, 95% CI 0.90–0.97) (Table 3).

Adjusted odds ratios for the different Robson groups were also estimated, apart from groups 6 to 9, which presented very small sample size. Women in groups 1+2 had a 12% (OR 0.88, 95% CI 0.83–0.93) lower chance of having a CS, with women in groups 3+4 experiencing 17% (OR 0.83, 95% CI 0.78–0.89). Women having premature babies (Group 10) presented a 22% lower chance, although wide CI were obtained due to the sample size (OR 0.78, 95% CI 0.68–0.90). Nevertheless, the reduction was not observed in every group, since Group 5 demonstrated no significant differences for any of the years (OR 0.95, 95% CI 0.87–1.03, 2017) (Table 3).

**Table 2. Association between caesarean section and socio-demographic and obstetric characteristics.**

| | Crude odd ratio | | | Adjusted odd ratio | | |
|---|---|---|---|---|---|---|
| | OR | 95% CI | p-value | OR | 95% CI | p-value |
| **Maternal age years** | | | | | | |
| <20 | 0.65 | 0.60–0.70 | <0.001 | 0.64 | 0.59–0.69 | <0.001 |
| 20–24.9 | 0.79 | 0.76–0.82 | <0.001 | 0.82 | 0.78–0.86 | <0.001 |
| 25–29.9 | 1 | | | 1 | | |
| 30–34.9 | 1.16 | 1.13–1.20 | <0.001 | 1.1 | 1.07–1.14 | <0.001 |
| 35–39.9 | 1.5 | 1.46–1.55 | <0.001 | 1.37 | 1.33–1.42 | <0.001 |
| ≥40 | 2.25 | 2.15–2.35 | <0.001 | 2 | 1.90–2.10 | <0.001 |
| **Nationality** | | | | | | |
| Spain | 1 | | | 1 | | |
| Rest of Europe and Central Asia | 0.84 | 0.80–0.88 | <0.001 | 0.86 | 0.82–0.90 | <0.001 |
| Middle East and North Africa | 0.71 | 0.69–0.73 | <0.001 | 0.77 | 0.74–0.80 | <0.001 |
| Latin America and Caribbean | 1.16 | 1.12–1.20 | <0.001 | 1.16 | 1.12–1.21 | <0.001 |
| East Asia and Pacific | 0.65 | 0.60–0.70 | <0.001 | 0.7 | 0.64–0.76 | <0.001 |
| Sub-Saharian Africa | 1.15 | 1.09–1.23 | <0.001 | 1.21 | 1.12–1.29 | <0.001 |
| South Asia | 1.19 | 1.13–1.27 | <0.001 | 1.15 | 1.07–1.22 | <0.001 |
| North America | 0.51 | 0.32–0.83 | 0.006 | 0.49 | 0.29–0.82 | 0.007 |
| **Socioeconomic status** | | | | | | |
| Disadvantaged population | 0.84 | 0.81–0.89 | <0.001 | 0.9 | 0.86–0.96 | <0.001 |
| Income <18.000 € | 1 | | | 1 | | |
| Income 18.000–100.000 € | 1.07 | 1.04–1.09 | <0.001 | 0.87 | 0.85–0.90 | <0.001 |
| Income >100.000 € | 0.73 | 0.46–1.18 | 0.002 | 0.47 | 0.27–0.80 | 0.006 |
| **Healthcare facility** | | | | | | |
| Level IA | 1 | | | 1 | | |
| Level IIA | 1.03 | 1.00–1.06 | 0.065 | 0.95 | 0.92–0.99 | 0.001 |
| Level IIB | 1.06 | 1.02–1.09 | 0.001 | 0.94 | 0.90–0.97 | <0.001 |
| Level IIIA | 1.2 | 1.17–1.23 | <0.001 | 1.09 | 1.05–1.12 | <0.001 |
| Level IIIB | 1.37 | 1.33–1.42 | <0.001 | 1.07 | 1.03–1.11 | <0.001 |
| **Pregnancy complication** | | | | | | |
| No | 1 | | | 1 | | |
| Yes | 1.43 | 1.41–1.46 | <0.001 | 1.44 | 1.41–1.47 | <0.001 |
| **Robson classification** | | | | | | |
| Group 1+2 | 1 | | | 1 | | |
| Group 3+4 | 0.38 | 0.37–0.39 | <0.001 | 0.36 | 0.35–0.37 | <0.001 |
| Group 5 | 3.87 | 3.75–4.00 | <0.001 | 3.55 | 3.43–3.67 | <0.001 |
| Group 6 | 141.44 | 107.94–185.34 | <0.001 | 145.07 | 110.67–190.16 | <0.001 |
| Group 7 | 78.92 | 64.44–96.66 | <0.001 | 78.29 | 63.89–95.94 | <0.001 |
| Group 8 | 5.53 | 5.18–5.91 | <0.001 | 4.91 | 4.59–5.26 | <0.001 |
| Group 9 | 86.62 | 59.38–126.36 | <0.001 | 74.67 | 51.14–109.03 | <0.001 |
| Group 10 | 1.75 | 1.67–1.83 | <0.001 | 1.59 | 1.52–1.66 | <0.001 |

It was thus established that fewer CS were performed in Catalonia between 2013 and 2017. Within the years that showed the greatest decrease (p-value <0.05) 2015, 2016, 2017 in CS, up to 2.319 CS (95% CI, 1399–3239) were avoided.

Model validation showed that the crude analysis which included all women was well calibrated and had a high grade of discrimination (calibration belt p-value: 0.915, ROC 0.75, 95% 0.74–0.75), however once the model was adjusted the calibration was dramatically affected (p-

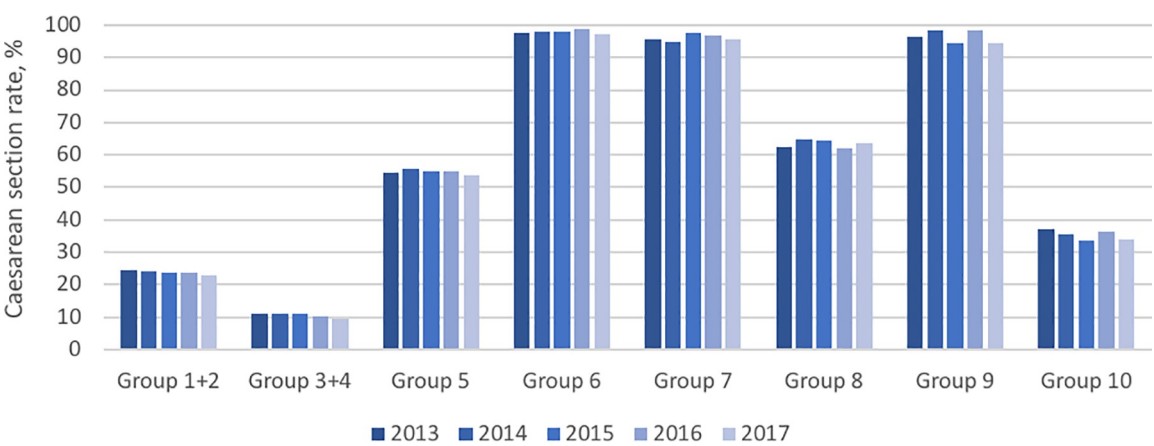

**Fig 2. Caesarean section rate.**

value 0.01) while the discriminant ability remained similar (ROC 0.77, IC 95% 0.76–0.77). The analysis of the different Robson groups showed good calibration but little discriminatory effect.

## Discussion

### Main findings

This study identified a significant decrease in CS rates in publicly funded hospitals in Catalonia between 2013 and 2017. When adjusted the CS rates for well-known risk factors, the reduction was even more pronounced. Considering that the characteristics of women are evolving into a higher risk profiles, this means that in equal conditions there are now less CS performed.

It is important to note that nulliparous (Group 1+2), multiparous (Group 3+4) and multiparous women with previous CS (Group 5) contributed most to the overall CS rate, suggesting that possibly are the groups that present larger margins for interventions on the aim of reducing final percentages of CS.

### Strengths

The most significant contribution of this study is the analysis of the individual level data from administrative registries. It enabled the analysis of CS rates for each group of Robson, rather

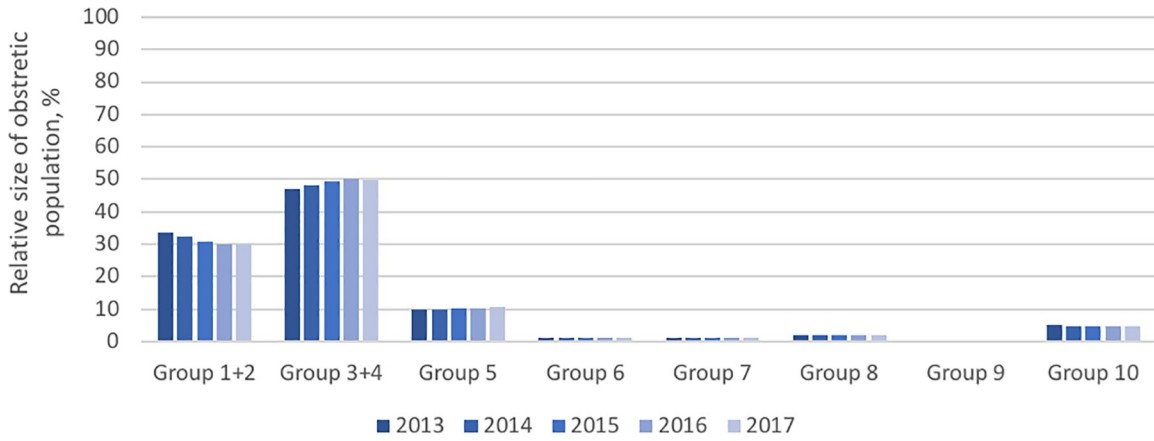

**Fig 3. Relative size of the obstetric population.**

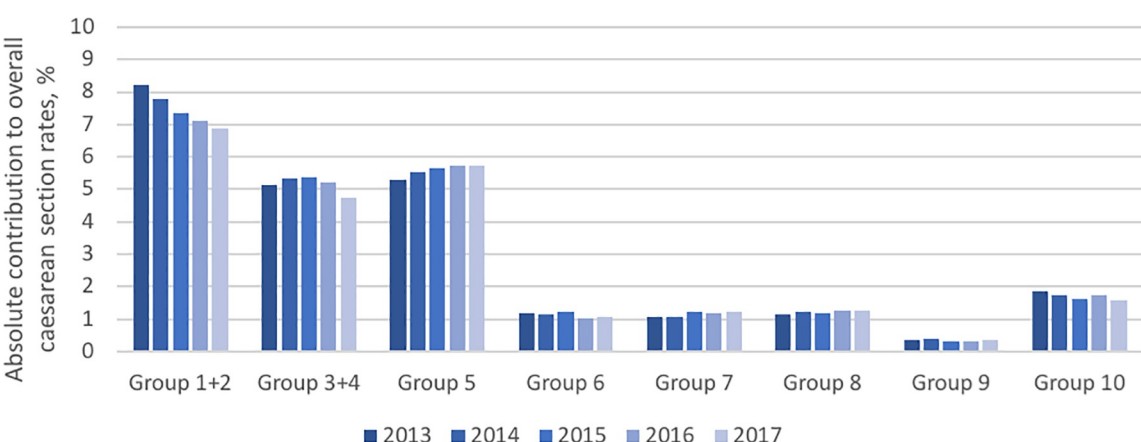

**Fig 4. Absolute contribution to overall caesarean section rates.**

than for the population, as simplified to a single number. Also, the large sample provided an important statistical power to the evidence shown and is innovative since includes adjustment variables not yet examined together in this type of research before.

A considerable difference in CS rates was identified between public and private settings in Catalonia [25]. The focus on deliveries in publicly funded hospitals excluded CS on demand, thus avoiding any possible bias.

To the best of the current researcher's knowledge, this is the first time the Robson classification has been implemented at a population level, within the Mediterranean countries, employing real world data and with consideration of so many confounders. The results could potentially be generalized to other Southern European countries with a similar healthcare framework.

## Limitations

The categorisation of the 10 groups of Robson was constrained by not having the variable onset of labour recorded, therefore a modified version of the Robson classification had to be created. Also, the registry does not collect some other variables which they had to be created ad hoc, i.e. foetal presentation, which was considered cephalic in the absence of a diagnosis of breech or transverse lie. When performing the crosscheck analysis with the measures for data collection quality suggested at the Robson Implementation Manual [32], the parameters in general showed consistency with those proposed. However, having to create the variable presentation ad hoc implied that not all of them were actually transverse, because CS for Group 9 did not end up being 100%. Also, creating variables ad hoc implied a lack of unclassified cases, which is considered another parameter to measure data collection quality.

The variable BMI had many omissions and therefore could not be included as an adjusting factor. However, the sensibility analysis showed a similar decrease in CS rates (S3 Table). The analysis of the missing data on the primary outcome revealed more nulliparous than multiparous women (S4 Table). However, in view of the small percentage (6%) it was believed that it did not imply deliberate avoidance by hospitals of reporting CS. Furthermore, it was considered that due to the large study sample, results would not have changed significantly. As well as the variable of "pregnancy complication" that since relies on the provider's clinical diagnosis, could be prone to bias.

This study sample tended to under-represent the highest socioeconomic category, due to private care, but in general, found no differences in CS rates between extreme socioeconomic

**Table 3. Time trends in caesarean section rates in all women and by groups of Robson.**

| | Calibration belt p-value | Area under the ROC curve | 2013△ | 2014 | | | 2015 | | | 2016 | | | 2017 | | |
|---|---|---|---|---|---|---|---|---|---|---|---|---|---|---|---|
| | | Area CI95% | CS rate CI95% | OR | CS rate CI95% | OR CI95% | CS rate CI95% | OR | OR CI95% | CS rate CI95% | OR | OR CI95% | CS rate CI95% | OR | OR CI95% |
| **Crude** | | | | | | | | | | | | | | | |
| All | 0.915 | 0.75 0.74–0.75 | 24.3 23.90–24.71 | 1 | 24.21 23.80–24.61 | 0.99 0.96–1.03 | 23.96 23.56–24.37 | 0.98 | 0.95–1.01 | 23.63 23.22–24.04 | 0.96 | 0.93–0.99* | 22.8 22.40–23.20 | 0.92 | 0.89–0.95*** |
| Robson 1+2 | 1 | 0.51 0.50–0.51 | 24.5 23.80–25.20 | 1 | 23.98 23.27–24.69 | 0.97 0.92–1.03 | 23.77 23.04–24.50 | 0.96 | 0.91–1.02 | 23.7 22.95–24.44 | 0.96 | 0.90–1.01 | 22.86 22.12–23.59 | 0.91 | 0.86–0.97*** |
| Robson 3+4 | 1 | 0.51 0.50–0.51 | 10.94 10.51–11.37 | 1 | 11.05 10.62–11.48 | 1.01 0.95–1.08 | 10.88 10.45–11.30 | 0.99 | 0.93–1.06 | 10.41 10.00–10.83 | 0.95 | 0.89–1.01 | 9.49 9.09–9.89 | 0.85 | 0.80–0.91** |
| Robson 5 | 1 | 0.51 0.50–0.51 | 54.35 52.84–55.85 | 1 | 55.86 54.36–57.37 | 1.06 0.98–1.16 | 55.04 53.56–56.53 | 1.03 | 0.94–1.12 | 54.89 53.40–56.37 | 1.02 | 0.94–1.11 | 53.58 52.12–55.04 | 0.97 | 0.89–1.06 |
| Robson 10 | 1 | 0.50 0.50,–0.51 | 37.2 35.14–39.25 | 1 | 35.54 33.47–37.61 | 0.93 0.82–1.06 | 33.7 31.63–35.78 | 0.86 | 0.76–0.98* | 36.2 34.08–38.32 | 0.96 | 0.84–1.09 | 33.89 31.77–36.00 | 0.87 | 0.76–0.98* |
| **Adjusted** | | | | | | | | | | | | | | | |
| All† | 0.01 | 0.77 0.76–0.77 | 24.56 24.20–24.92 | 1 | 24.29 23.93–24.65 | 0.98 0.95–1.02 | 23.9 23.54–24.26 | 0.96 | 0.92–0.99* | 23.59 23.22–23.95 | 0.93 | 0.90–0.97*** | 22.56 22.20–22.92 | 0.87 | 0.84–0.90*** |
| Robson 1+2• | 0.17 | 0.59 0.59–0.60 | 24.92 24.23–25.62 | 1 | 24 23.29–24.70 | 0.95 0.90–1.00 | 23.72 22.99–24.44 | 0.93 | 0.88–0.99* | 23.52 22.78–24.26 | 0.92 | 0.87–0.98** | 22.61 21.89–23.33 | 0.88 | 0.83–0.93*** |
| Robson 3+4• | 0.86 | 0.59 0.58–0.59 | 11.14 10.70–11.57 | 1 | 11.05 10.62–11.48 | 0.99 0.93–1.05 | 10.81 10.39–11.23 | 0.97 | 0.91–1.03 | 10.33 9.92–10.75 | 0.92 | 0.86–0.98** | 9.45 9.05–9.85 | 0.83 | 0.78–0.89*** |
| Robson 5• | 0.74 | 0.59 0.58–0.59 | 54.64 53.15–56.13 | 1 | 55.97 54.48–57.46 | 1.06 0.97–1.15 | 55.15 53.68–56.62 | 1.02 | 0.94–1.11 | 54.69 53.22–56.17 | 1 | 0.92–1.09 | 53.29 51.84–54.74 | 0.95 | 0.87–1.03 |
| Robson 10• | 0.76 | 0.58 0.57–0.58 | 37.97 36.01–39.92 | 1 | 36.12 34.15–38.09 | 0.91 0.80–1.05 | 34.11 32.13–36.10 | 0.83 | 0.73–0.95** | 35.47 33.48–37.47 | 0.89 | 0.77–1.01 | 32.9 30.92–34.88 | 0.78 | 0.68–0.90*** |

△Reference group

† Adjusted for maternal age, nationality, socioeconomic status, healthcare facility, pregnancy complication, delivery year and modified Robson classification.

• Adjusted for maternal age, nationality, socioeconomic status, healthcare facility, pregnancy complication, delivery year and modified Robson classification.

‾Goodness of fit assuming polynomial relationship

*p-value <0.05

**p-value <0.01

***p-value<0.001

levels. The numbers suggested that more Catalan nulliparous women attended the private sector, as this sample was constituted of 35.81% nulliparous and 64.19% multiparous, while national statistics showed about 50% [27].

Regarding the statistical analysis, the purpose of the study hampered the possibility of applying a multi-level approach. Although it is realistic to think that same women could have delivered more than once within the study period, the magnitude of the population sample should still provide relevant and valuable results.

## Interpretation

Following adjustment for individual risk factors, CS rates decreased in Catalonia between 2013 and 2017. However, comparison with other countries and aiming for optimal rates suggests scope for improvement [26,35].

Our study founds similar patterns to that observed in other European countries, groups 1 to 5 are the once contributing the most to the final CS rate [35]. However, attention should be paid to the reduced proportion of nulliparous women in the study sample, since this made the highest contribution to the overall CS rate and would had been higher in the absence of private care.

In addition, the evidence suggests that optimising CS rates requires focusing attention on groups 1 to 5. However, the ability to prioritize interventions as a result of our study is compromised by the inability to determine the onset of labour.

CS rates in Group 5 (i.e. women with previous CS) reflected the obstetric practice of previous years. It is particularly linked to CS performed on nulliparous women, and will therefore, if this continues to decrease, be reflected in Group 5. Also suggests that trial of labour following CS (TOLAC) should be prioritized in favour of optimum levels [36].

Rates for women with breech presentations remained the same in our study (groups 6 and 7). Following the Term Breech Trials results, CS for breech rapidly increased worldwide [37]. Spanish guidelines have recently upheld considering vaginal birth acceptable under some circumstances [38,39], with some hospitals in Catalonia now resuming them, although it will take time for numbers to reflect any change. Facilitating the external cephalic version, is in any case the preferred option, also for transverse presentations (group 9) [40–42].

The reason behind the overall decrease remains unclear. The protocol to promote natural birth [43], the adequacy of indications, the creation of an adjusted index for CS [44] or growing evidence of the risks that implies the CS [45] could only be some of the reasons behind. In addition, the fourth-wave of feminism, international organisations on birth-rights together with local organisations or growing eco on the media regarding obstetric violence, could have also influenced this reduction [46,47].

Furthermore, the application of systematic Robson classification has been suggested to contribute on the decrease of CS rates. In line with this, interventions and indications should also be audited. CS rates can be safely reduced by applying multifaceted strategies combining audit and feedback, implementing guidelines on mandatory second opinion or educating physicians by local opinion leaders [1,48]. In addition, midwife-led continuity models of care have also proved beneficial, including reducing CS for low risk women [49–51]. In Catalonia, one out of the seven projected midwifery-led units is already functioning. However, midwife ratios in Spain continue to be one of the lowest across European countries [52].

This analysis hopes to provide useful information to the public, public health experts and health professionals, however, does not intend to replace in any case the original Robson classification. It should only be a temporary tool until all needed variables are systematically recorded.

## Conclusion

This study found a significant decrease in CS rates in Catalonia between 2013–2017. Provides evidence that group 1+2, 3+4 and 10 are the groups that have shown the highest reduction after adjusting for confounders and suggests retrospective overutilization of CS and percentages of (in)adequacy in the past. Special attention should be paid to groups 1 to 5 since they imply the biggest contributors to the overall CS rates, and any reduction would imply a considerable reduction in the total rate.

The reasons behind remains unclear and further efforts should be made as rates remain above optimum levels. Including the Robson classification as a systematic way to analyse CS could be very useful to compare, assess and analyse data and prioritize, however all variables including onset of labour are necessary. Hence, policy makers should give urgent attention to the collection of perinatal data.

## Supporting information

**S1 Checklist. STROBE checklist.**
(DOCX)

**S1 Table. Diagnostic codes included in the variable pregnancy complication.**
(DOCX)

**S2 Table. Bivariate analysis of socio-demographic and obstetric characteristics and type of delivery.**
(DOCX)

**S3 Table. Time trends in caesarean section rates including the variable BMI by groups of Robson.**
(DOCX)

**S4 Table. Socio-demographic and obstetric characteristics of study population by type of delivery.**
(DOCX)

## Acknowledgments

The authors thank all the stimulating discussions and guidance from the following: Alba Oliver-Parra (UPF), Cristina Colls-Guerra (AQuAS), Cari Almazán-Sáez (AQuAS) and Mercè Armelles-Sebastiá (Maternal and Child Health Services, Catalonian Public Health Agency, Department of Health).

## Author Contributions

**Conceptualization:** Garazi Carrillo-Aguirre, Albert Dalmau-Bueno.

**Data curation:** Garazi Carrillo-Aguirre, Albert Dalmau-Bueno.

**Formal analysis:** Garazi Carrillo-Aguirre.

**Methodology:** Garazi Carrillo-Aguirre, Albert Dalmau-Bueno, Carlos Campillo-Artero.

**Project administration:** Albert Dalmau-Bueno, Anna García-Altés.

**Writing – original draft:** Garazi Carrillo-Aguirre.

**Writing – review & editing:** Garazi Carrillo-Aguirre, Albert Dalmau-Bueno, Carlos Campillo-Artero, Anna García-Altés.

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
