## [Decision Letter · Decision Letter 0]

17 Feb 2020

PONE-D-20-00730

Caesarean section trends in Catalonia between 2013 and 2017 based on the Robson classification system: a cross-sectional study.

PLOS ONE

Dear Dr. García-Altés,

Thank you for submitting your manuscript to PLOS ONE. After careful consideration, we feel that it has merit but does not fully meet PLOS ONE’s publication criteria as it currently stands. Therefore, we invite you to submit a revised version of the manuscript that addresses the points raised during the review process.

We would appreciate receiving your revised manuscript by Apr 02 2020 11:59PM. To enhance the reproducibility of your results, we recommend that if applicable you deposit your laboratory protocols in protocols.io, where a protocol can be assigned its own identifier (DOI) such that it can be cited independently in the future. For instructions see: http://journals.plos.org/plosone/s/submission-guidelines#loc-laboratory-protocols

We look forward to receiving your revised manuscript.

Kind regards,

Calistus Wilunda, DrPH

Academic Editor

PLOS ONE

Additional Editor Comments (if provided):

Having a large sample size does not necessarily address the potential for bias as claimed on page 15/16. Please drop/revise this statement. 

Journal Requirements:

2. In ethics statement in the manuscript and in the online submission form, please provide additional information about the databases used in your retrospective study. Specifically, please ensure that you have discussed whether all data were fully anonymized before you accessed them and/or whether the IRB or ethics committee waived the requirement for informed consent. If patients provided informed written consent to have their data used in research, please include this information.

Reviewers' comments:

Reviewer's Responses to Questions

**Comments to the Author**

1. Is the manuscript technically sound, and do the data support the conclusions?

Reviewer #1: Partly

Reviewer #2: Yes

Reviewer #3: Partly

Reviewer #4: Yes

2. Has the statistical analysis been performed appropriately and rigorously? 

Reviewer #1: No

Reviewer #2: Yes

Reviewer #3: Yes

Reviewer #4: I Don't Know

3. Have the authors made all data underlying the findings in their manuscript fully available?

Reviewer #1: Yes

Reviewer #2: Yes

Reviewer #3: Yes

Reviewer #4: No

4. Is the manuscript presented in an intelligible fashion and written in standard English?

Reviewer #1: Yes

Reviewer #2: Yes

Reviewer #3: Yes

Reviewer #4: Yes

5. Review Comments to the Author

Reviewer #1: This is a retrospective cross-sectional study of over 200,000 deliveries over a 5 year period in Catalonia. It is overall well written and as the authors state, the Robson classification is a very useful system to compare, assess and analyze data in relation to Caesarean delivery.

Unfortunately, there are major flaws with the methodology of this work.

1) Although this study is not representative of the entire population, almost 10% of data is incomplete and therefore omitted which is of concern.

2) The CD rate in Group 9 should be 100% and is a measure of data collection quality, with anything less than this suggesting errors in data collection.

3) By combining group 1 and 2 (and 3 and 4), analysis of CS in spontaneous labour and induction of labour in nulliparous and multiparous women cannot be undertaken.

Reviewer #2: Background:

Line 59 to 64

1. Monitoring of CS rate started in 1990 and different published National Health Plans have included the objective of CS 60 reduction [14–18].

Need little bit more explanation what strategies are in place to monitor CS rate? is it only by rate/ by indication? If it is by rate how the CS target for a specific hospital has been set.

2. However, even though having had also implemented several protocols [19–22], it is 61 difficult to estimate their impact.

What protocols and why you consider difficulties present to estimate the impact.

3. Rates have ranged between 22% and 32%, with considerable 62 differences between the public and the private sectors, i.e. 22.3% to 35.9% for the year 2017 [23].

Does the protocols are only applicable for public funded hospitals. Do you know why the CS rates vary in private hospitals. These are for profit or including bothe for profit & NGO funded?

Maternal and neonatal mortality ratio has remained very low

It is better to mention MMR/ NMR in numerical term rather simply saying very low. If you want to co-relate the rate with CS trend more appropriate will be to explain the trend of MMR & NMR

Methods:

1. How the variable socio economic status constructed in the data bases are essential to state out

2. Do not get the point hospital level! how the least and most defined within the data bases

3. Maternal complications: I would rather say it pregnancy risk factors

4. Is the BMI denotes pre-pregnancy BMI?

5. Line 147: The aim of the study was to analyse CS trends and, in any case, pretended to be a prognostic model to predict the risk of CS.

This aim has not been mentioned in your introduction. Better to say determinants of CS births rather prognostic model

6. I do not heard about STATA 14.2 . please add reference for this line

Should not it be 14/15/16

Result:

Table 1 need heading for each column

Table 2: better to put the reference value at the top row for each variable

Discussion:

1. Line 281 However, comparison with other countries and aiming for optimal rates suggests scope for improvement [32]

What is the expected optimum rate?

2. High CS Rate in group 1&2 increases the chance of increase CS rate in group 5

3. I found the data set CMBD has the CS indication data. Why the author does not take this advantage to describe the CS rate according to indication? Then it would be more inclusive to obstetric determinants of CS

Reviewer #3: Thank you for the opportunity to review this paper. This is a well written manuscript that has design flaws which prevent it meeting its potential and delivering on its title. The main selling point seems to be the use of the Robson Ten Group Classification System (TGCS) however full description of the TGCS is impossible due to a lack of data relating to onset of labour. Similarly the paper falls between the two stools of using the TGCS and also looking at factors which have nothing to do with obstetric characteristics such as socio-economic status.

These factors lead to the manuscript being confused and crowded with unnecessary data and figures which make it difficult to digest.

I would suggest removing the Robson TGCS data entirely from the manuscript as without a basic piece of information such as onset of labour the TGCS is dysfunctional and does not provide meaningful information.

The introduction section is well written and clear.

Methods are confused due to not being able to clearly define the TGCS, the sample size however is impressive and the use of three separate data sources would suggest that the authors have gone as far as possible to gather all the available data possible.

The results section has issues. The inability to define the TGCS completely in the sample leads to use of a modified system which leads to two Groups being enormous compared to the other six and this leads to poorly conceived figures. The TGCS is "shoe-horned" into the manuscript and does not add anything to the overall message. Maybe the paper would be better served by looking at purely patients demographics and characteristics and how they contributed to the decrease in the rate of CS. Use of the TGCS in Table 2 is particularly unnecessary. Using Group 1+2 simply cannot be compared to other groups in the system. The whole point of the TGCS is that the groups mutually exclusive and unrelated to each other. For example saying that a patients Odds Ratio of having a CS in Group 6 (Nulliparous Breech) vs Group 1+2 = 141 adds absolutely nothing the papers message. The inclusion of p-values in a sample size such as this is also not necessary as they are almost routinely <0.0001.

The figures are also rendered useless by the use of the modified TGCS. Figure 2 is particularly not worthwhile.

While I appreciate the authors have spent much time and effort in producing this paper I feel it needs significant work and the TGCS cannot be the centre piece and probably should be excluded and definitely should not be mentioned in the title.

Reviewer #4: This is a manuscript that reports trends on the use of caesarean section and determinants using the Robson classification system retrospectively from 2013 to 2017. The analysis includes all deliveries in the public system in Catalonia (44 facilities). The Robson system is recognized to allow studying perinatal events (originally, caesarean section rates) in more homogeneous groups of women in whom to focus interventions. The classification has been endorsed by several international normative bodies such as WHO, FIGO and EBCOG. In the current international context of increasing caesarean section rates worldwide and the debate on the overuse with medical and non-medical factors underlying the use, this is an interesting and relevant manuscript to understand trends in groups of women. Importantly, it includes all hospitals and thus all levels within the public health system.

Some consideration for improving the clarity of the manuscript follow:

• Population: it is not fully clear in the methods the issue of missing data. In page 4 line 80, authors state that 9.1% of the women had missing data. It is, nonetheless, not clear to me the inclusion/exclusion criteria. What were the variables that were considered essential for inclusion of the women in the analysis? The variables necessary for the Robson classification? All variables used in the analysis? Please clarify in the Methods section. Authors could considered a flowchart showing the process of the exclusion of women to arrive to the final sample size of their analysis (this could be shown in the first paragraph of the results).

• Population: Consider adding the proportion of births occurring in the public health system among all births in Catalonia as an indication for the reader of the coverage of the analysis.

• Data sources: one of the strengths of the analysis and manuscript is the richness of the data due to the availability and inclusion of multiple data sources in the analysis. Were there any inconsistencies found between the data in the three databases? If so, how were they resolved? For example, if I understand correctly, “parity” seem to be available through the CMBD and the E-CAP. Was there any hierarchy from where to select the data?

• Variables: page 6 line 117-122: Some complications that may be important determinants of mode of birth are not in the list. Eclampsia is listed in Table S1 but I wonder about others such as infections of any type, haemorrhage or stillbirths. Was there no data on these complications available?

• Variables: the variable “socioeconomic status” seems to be “income”, correct? If so, consider calling it “income”. Also, I suggest that authors clarify/define the categories. Particularly, it is not clear what “disadvantaged populations” are and what marks the difference between the group with income <18,000 and the group under disadvantage populations (assuming they are mutually exclusive).

• Results, Page 7 line 156 and 157: authors describe decreases and increases as “reasonable”. Please clarify what “reasonable” is or delete. I think it will be appropriate to mention in the results those changes that are statistically significant.

• Results, page 9 line 168: “… particularly for those over thirty-five, who presented a 37% higher change….” According to Table 2, the 37% higher risk is for women 35-39.9 years of age not for all those above 35 years.

• Results, page 11 line 194-196: “Within the population sample, the majority of women carried single babies at term and belonged to 195 groups 1+2 and 3+4 (31.45% and 48.88%, respectively)….”. It should be added “in cephalic presentation”.

• Discussion, page 15 line 256-257: authors acknowledged under the limitations that cephalic presentation was diagnosed in the absence of a direct registration of a diagnoses of breech or transverse lie. I think this information should come also under the Methods section with all other definitions or methodological particularities of the variables.

• Discussion, page 17 line 289-292: authors discussed their CS rate in Robson Group 5 which is 54.7%. On the basis of available evidence, this is a relatively low CS rate for this group. I wonder if authors could discussed if there is any specific policy for TOLAC in Catalonia and if they think this CS rate in Group 5 is homogeneous across the region or, on the contrary, is likely to suffer great variations between settings. Although it is not validated, the Robson Classification Implementation Manual published by WHO (https://www.who.int/reproductivehealth/publications/maternal_perinatal_health/robson-classification/en/) initially suggests that “Rates of 50-60% in Group 5 are considered appropriate provided you have good maternal and perinatal outcome”.

• Discussion, page 17 lines 293-298: authors discuss the CS rates in breech presentations acknowledging that the use of CS in breeches has been driven by the results of the Term Breech Trial. Important to note that the authors of the 2-year trial follow-up noted that ‘‘planned caesarean delivery is not associated with a reduction in risk of death or neurodevelopmental delay in children at 2 years of age’’ [Reference: Whyte H, Hannah ME, Saigal S, et al. Outcomes of children at 2 years after planned cesarean birth versus planned vaginal birth for breech presentation at term: the International Randomized Term Breech Trial. Am J Obstet Gynecol 2004; 191: 864–71.]

• Discussion, page 17 line 299-303; authors discuss their hypothesis and the potential reasons behind the decrease of CS in Catalonia. In contemporary societies where fear of litigation and fear of damage reputations are significant factors underpinning the increase of CS, a 1.5 percent points of significant decrease is impressive, particularly at population level and in Robson Groups 1 through 4. I wonder if the authors would like to add in the discussion about this point or if they know of any on-going efforts in Catalonia to mitigate the fear among healthcare professionals. This would be very useful for other countries under similar circumstances.

• Discussion: Of interest is the reverting pattern shown on CS rates in Catalonia with regard to income level. The normal pattern seen in the literature tends to show higher CS rates in the wealthier women whereas this study shows the contrary. I understand that most of the wealthiest women would be giving birth in the private sector and thus, if incorporated in this analysis, they could have increased the CS rate of the wealthier groups in this paper. Do authors think that this is the only explanation for this pattern in Catalonia or they think they may be something else underlying the pattern?

6. PLOS authors have the option to publish the peer review history of their article (what does this mean?). If published, this will include your full peer review and any attached files.

Reviewer #1: No

Reviewer #2: Yes: Tahmina Begum

Reviewer #3: No

Reviewer #4: No

---

## [Author Response · Author response to Decision Letter 0]

1 May 2020

Dear Dr. Wilunda,

I would like to thank you for your attention and the reviewers for their evaluation on our paper. We really appreciate the valuable comments, which have provided insights that helped improve the paper. We tried to follow all suggestions and changed the manuscript accordingly. Positive comments have not been answered, although highly appreciated. Please find below a structured table answering all the comments thoroughly. 

Thank you very much, looking forward to hearing from you.

Yours sincerely,

Anna García-Altés, as author for correspondence

*Changes made: Pages and lines numbers, refer to the document “Revised_Manuscript_with_Track_Changes”

Editors’ comments Authors’ response Changes made

Having a large sample size does not necessarily address the potential for bias as claimed on page 15/16. Please drop/revise this statement. Completely agree. We have removed the statement. Page 14/15. It has been rephrased due to another comment. 

1. Please ensure that your manuscript meets PLOS ONE's style requirements, including those for file naming. The requirements have been met. -

2. In ethics statement in the manuscript and in the online submission form, please provide additional information about the databases used in your retrospective study. Specifically, please ensure that you have discussed whether all data were fully anonymized before you accessed them and/or whether the IRB or ethics committee waived the requirement for informed consent. If patients provided informed written consent to have their data used in research, please include this information. A governmental structure called PADRIS exists within the Catalan healthcare administration. PADRIS is an analytical data programme for research and innovation in health. The purpose of the structure is to make data available to scientific communities to promote research, innovation, and evaluation in health, with the aim of reusing and exchanging the data generated by the health system in accordance with the legal and regulatory framework. The programme PADRIS ensures that all data made available to researchers is fully anonymised. As workers within the Catalan healthcare administration, we are provided data by PADRIS, and therefore are never given access to identifiable information. Thus, the study did not involve any data collection, requiring neither human participants nor patient consent. For that reason, and due to the use of existing anonymised data for research, the study was exempt from institutional review committee approval. It is the standard way of proceeding in the healthcare administration to systematically check the quality of the healthcare providers in our context. -

3. We note that you have indicated that data from this study are available upon request. PLOS only allows data to be available upon request if there are legal or ethical restrictions on sharing data publicly. a) If there are ethical or legal restrictions on sharing a de-identified data set, please explain them in detail (e.g., data contain potentially identifying or sensitive patient information) and who has imposed them (e.g., an ethics committee). Please also provide contact information for a data access committee, ethics committee, or other institutional body to which data requests may be sent. According to the legal framework by the Catalan Government regarding data from PADRIS, anonymised data sets are only available to Catalan universities, Catalan research centres, and Catalan healthcare centers, by means of signing a written statement. Therefore, for legal reasons, data will potentially not be available for PLOS. For further information, you can visit: http://aquas.gencat.cat/ca/ambits/analitica-dades/padris/.

-

Reviewer #1 Author response Changes made

1) Although this study is not representative of the entire population, almost 10% of data is incomplete and therefore omitted which is of concern. The study is not representative of the entire population because private hospitals do not have a legal obligation to report data to the authorities and governmental registries in a traceable way. Therefore, it would not be possible to gain access to this information in any case. However, we consider the data to be representative of the women that attend the publicly founded hospitals in Catalonia (70% of the total deliveries in Catalonia are attended in public hospitals). 

In addition, there is no international consensus nor a specific % of missing data to be considered acceptable in cross-sectional population-based studies. However, we have followed the STROBE checklist, on which there is consensus; besides that, we consider 10% more than acceptable for this type of studies – we have performed a sensibility analysis which is available as supporting information. Furthermore, we consider it is also worth noting the total population of the study, n= 231.020. N/A.

2) The CD rate in Group 9 should be 100% and is a measure of data collection quality, with anything less than this suggesting errors in data collection. We agree it is a measure of quality assurance. However, we would like to clarify that it is not due to any data collection error, but to the lack of notification of the variable transverse in the national registry. 

Creating the variable transverse ad hoc was the most feasible option, hence, considering transverse in the absence of the variable cephalic or breech presentation. Unfortunately, this method had certain consequences; specifically, as already acknowledged in the limitations, that not 100% of cases ended up transverse in reality. We wanted to be transparent and honest with our results and acknowledge this limitation, not only because we believe it is how science and research should be, but also because it can help future researchers and health systems stakeholders. 

In addition, we would like to emphasise that this is not a hospital database, internally designed to analyse exclusively sexual and reproductive health issues with specifically adjusted ad hoc variables, but a national registry consisted of population-scale data. Page 15, line 278. When performing the crosscheck analysis with the measures for data collection quality suggested at the Robson Implementation Manual [32], the parameters in general showed consistency with those proposed. However, having to create the variable presentation ad hoc implied that not all of them were actually transverse, because CS for Group 9 did not end up being 100%. Also, creating variables ad hoc implied a lack of unclassified cases, which is considered another parameter to measure data collection quality.

3) By combining group 1 and 2 (and 3 and 4), analysis of CS in spontaneous labour and induction of labour in nulliparous and multiparous women cannot be undertaken. As stated in the scientific paper published by PLOS ONE “A Systematic Review of the Robson Classification for Caesarean Section: What Works, Doesn’t Work and How to Improve It”, the ten groups constitute the backbone of the Robson classification; however, they also find that many studies subdivide or merge Robson groups. In our case, the CMBD national registry is the most powerful registry of individual data at a national level in health science. We believe that utilising the CMBD for the analysis of CS according to Robson Classification is an opportunity of great calibre, which should not be eradicated for not having the 10 groups. 

Also, we believe that our study demonstrates to other countries and healthcare systems that it is worth undertaking the analysis of CS rates according to Robson Classification (as endorsed and recommended by the WHO, FIGO…), even if they do not utilise all the variables described in the implementation manual. On the one hand, it is evident that an interesting discussion arose from the results and, on the other hand, approaching and facing the challenges provides an opportunity for improvement. This analysis has given us the best opportunity to recognise the weaknesses and flaws of the registry in terms of reproductive health. It has highlighted the gaps in data collection that will allow us to address the relevant stakeholders with sufficient evidence for it to be improved and action taken. We understand this is the best possible contribution in terms of the future and the best possible contribution to maternal health. N/A. 

Reviewer #2 Author response Changes made

Line 59 to 64 

1. Monitoring of CS rate started in 1990 and different published National Health Plans have included the objective of CS 60 reduction [14–18].

- Need little bit more explanation what strategies are in place to monitor CS rate? Is it only by rate/ by indication? 

-If it is by rate how the CS target for a specific hospital has been set. Thank you for your comment. In Catalonia, for many years the monitoring of CS rate has consisted of the analysis of a global single CS rate recorded at the CMBD, and also as a single percentage of CS rate per hospital.

In addition, for more than 5 years now, the Health Evaluation and Quality Agency of Catalonia (AQuAS) has reported the CS rate per hospital, adjusted according to an index. The variables included in the index adjustment are as follows: single or multiple pregnancy, previous CS, complications during labour, gestation and maternal conditions or complications. 

As in the Robson Classification, this index tries, by considering obstetric characteristics and intrinsic differences in hospital factors and infrastructures, to provide a framework to compare CS rate between hospitals.

As far as we know, there is no monitoring of CS by indication and there is no specific target CS rate for each hospital. No changes made. 

2. However, even though having had also implemented several protocols [19–22], it is 61 difficult to estimate their impact.

What protocols and why you consider difficulties present to estimate the impact. The protocols and guidelines implemented in Spain and Catalonia have been generic and have included only general recommendations about care during pregnancy and labour. Furthermore, the reports published to date have only compared CS rates among hospitals. In essence, there has not been any specific evaluation of the implementation or the impact of such documents. 

We have considered your comment and changed the text to provide further clarification. Page 3, line 60. However, despite having also implemented several protocols and guidelines in regards to care during pregnancy and delivery and the publication of official reports comparing CS rates between hospitals [19–24], their impact is unknown due to a lack of exhaustive evaluation assessments.

3. Rates have ranged between 22% and 32%, with considerable differences between the public and the private sectors, i.e. 22.3% to 35.9% for the year 2017 [23].

Does the protocols are only applicable for public funded hospitals. Do you know why the CS rates vary in private hospitals. These are for profit or including bothe for profit & NGO funded? In Catalonia, there are publicly funded hospitals and private hospitals. Private hospitals are for-profit healthcare organisations and there are no NGO-funded hospitals providing maternity care.

In regards to protocols and guidelines, these are meant to be for everyone; however, private hospitals tend to do things differently on occasion (i.e., if the protocols suggests three scans during pregnancy, a private obstetrician might offer more scans, one per appointment for instance). 

The reasons for the higher rates of CS in the private hospitals are based on speculation only, since they have not been studied. However, the rates have been well documented over a long period and the pattern is similar across the different Autonomous Communities in Spain. Some of the reasons behind might be: maternal request, fear of litigation, more medicalisation, scheduling inductions/CS according to obstetrician’s/woman’s convenience, or financial incentives. No changes made. 

4. Maternal and neonatal mortality ratio has remained very low

It is better to mention MMR/ NMR in numerical term rather simply saying very low. If you want to co-relate the rate with CS trend more appropriate will be to explain the trend of MMR & NMR Thank you for the comment. We do not currently have access to the trends of MMR and NMR for Catalonia for the years 2013–2017. However, considering they are very low with no significant changes between the years, we have included the average to give an idea. (They are so low that they do not reflect any change compared to CS rate). Page 3, line 65: (MMR: 3.1, 2010-2014 [26] and NMR: 1.67, 2014-2017 [27]).

Methods: 

1. How the variable socio economic status constructed in the data bases are essential to state out Thank you very much for pointing this out; this change will improve the reader’s understanding. 

 Page 6, line 122. The variable socioeconomic status is a set variable within the database that arises through pharmaceutical co-pay estimations. There are four co-payment groups: 1) those exempt from co-payment, (disadvantaged population; individuals receiving some form of universal pension scheme, those who no longer receive unemployment allowance, or those who no longer receive unemployment benefit and do not qualify for unemployment allowance); 2) those with annual earnings of less than 18,000€; 3) those with annual earnings between 18,000€ and 100,000€; and 4) those with annual earnings over 100,000€. This is predetermined by the system and does not allow further disaggregation. 

2. Do not get the point hospital level! how the least and most defined within the data bases Thank you for your comment. As mentioned in the manuscript, the hospital level follows an order of complexity. Level I hospitals provide assistance to low risk and medium risk pregnancies (i.e., premature babies > 35-36w without complications). Level IIA, low risk and high-risk pregnancies (i.e., premature babies >32w.). Level IIB low risk and high risk (i.e., premature babies > 28w). Level IIIA, low risk and high risk (i.e., premature babies of any gestation). Level IIIB low and high risk (i.e., premature babies of any gestation that will require super-specialists).

We have now referenced in the manuscript an official document from the Government of Catalonia that describes the hospital levels. It gives details about the resources that should be available at each level. We have also changed the statement in the manuscript to provide further clarification. Page 6, line 130. The hospital level follows an order of complexity, based on the care that the pregnant woman and her baby might require. Thus, even though all hospitals will be caring for women with low risk pregnancies, the more complex the pregnancy becomes the higher the level of the hospital in which she will be cared for. Level IA being the less medicalised hospitals, and IIIB the most, offering medical attention from other specialities if required [34].

3. Maternal complications: I would rather say it pregnancy risk factors We appreciate the advice but we prefer the concept “maternal complications”. We understand risk factors as something that increases a person’s chances of developing a disease or having a complication; however, in this case the condition is already present. In a hospital environment in the UK, for instance, it is common to talk about maternal complications, and we have also found sound scientific evidence using this terminology. No changes made. 

4. Is the BMI denotes pre-pregnancy BMI? Thank you for the comment. From clinical experience and according to the current guidelines, we presume it is pre-pregnancy or beginning of pregnancy BMI, as most of the time it is recorded at the booking appointment (guidelines suggest doing so at booking appointment). However, we cannot be 100% sure of this, as the computer system does offer the healthcare professional the option to record it further along in the pregnancy. No changes made.

5. Line 147: The aim of the study was to analyse CS trends and, in any case, pretended to be a prognostic model to predict the risk of CS.

This aim has not been mentioned in your introduction. Better to say determinants of CS births rather prognostic model Thank you for pointing this out, it was a grammatical mistake. The sentence has been appropriately changed. Page 8, line 165. The aim of the study was to analyse CS trends and the intention was not to build a predictive model for CS, although the possibility was explored.

6. I do not heard about STATA 14.2 . please add reference for this line

Should not it be 14/15/16 Stata 14.2 is a version of Stata 14. Therefore, I have changed the reference to Stata 14 in the text. Page 7, line 167. Stata software version 14.

Result: 

Table 1 need heading for each column We have included the word “year” at the top of each column for clarification. Year 2013

Year 2014

Year 2015

Year 2016

Year 2017

Table 2: better to put the reference value at the top row for each variable We appreciate the suggestion, but we have considered another criterion. For the variable maternal age, for instance, we did not want to consider the reference value <20 years, because it is very uncommon in our country to have babies below 20. By contrast, for the variable socioeconomic status, we considered income <18,000€ because the majority of women in the study belong to this group. No changes made. 

Discussion: 

1. Line 281 However, comparison with other countries and aiming for optimal rates suggests scope for improvement [32]

What is the expected optimum rate? Thank you for your comment. As stated in the introduction, recent studies have suggested optimal CS rates to be between 10–20% of all births (considering maternal and neonatal mortality). Copied from the introduction: In places where CS is universally available, though, optimal rates should be expected (i.e., Catalonia). Not without controversy, recent studies have suggested rates between 10–20% of all births [6–9].

In addition, a new reference has been included, European Perinatal health Report, where all CS rates for each European country are available. Reference added: European Perinatal Health Report. 

2. High CS Rate in group 1&2 increases the chance of increase CS rate in group 5 We presume this comment is regarding the paragraph starting in line 314: CS rates in Group 5 (i.e., women with previous CS) reflected the obstetric practice of previous years. It is particularly linked to CS performed on nulliparous women, and will, therefore, if this continues to decrease, be reflected in Group 5.

This statement is linked with the interpretation in the Robson Classification Implementation Manual: “Look at the absolute contribution of Group 5 to the overall CS rate: if it is very high, this may indicate that in previous years, CS rates in Groups 1 and 2 have been high and it is worth exploring further”. No changes have been made. 

3. I found the data set CMBD has the CS indication data. Why the author does not take this advantage to describe the CS rate according to indication? Then it would be more inclusive to obstetric determinants of CS We did explore this option, but unfortunately we could not include it mainly because of two reasons: 

- A large % of women without the indication recorded. 

- The indication of the CS is presumed to be the first diagnosis reported to the registry. However, there is often more than one diagnosis that could potentially be the reason for the CS reported in the registry. In view of the lack of accuracy, we decided to discard it. N/A. 

Reviewer #3 Author response Changes made

I would suggest removing the Robson TGCS data entirely from the manuscript as without a basic piece of information such as onset of labour the TGCS is dysfunctional and does not provide meaningful information. As mentioned above, we strongly believe that the analysis of the Robson Classification is important and meaningful in our context. We do not consider it dysfunctional in any case for not having the 10 groups. We believe that the results and the discussion withdrawn from the study are potentially very valuable to policymakers, public health experts or health professionals like obstetricians and midwives. 

Since the Robson Classification was presented in 2001, many countries have undertaken analysis of CS according to it. It is endorsed by the WHO, FIGO and EBCOG among others and, since 2015, the WHO has proposed the Robson Classification system as a global standard for assessing, monitoring, and comparing caesarean section rates within healthcare facilities over time, and between facilities. 

Until now, neither the Spanish nor the Catalan governments have undertaken or approximated the analysis of CS according to the Robson Classification. We therefore consider this a fantastic opportunity as a first approach to initiate the right pathway in the analysis of CS. N/A.

The results section has issues. The inability to define the TGCS completely in the sample leads to use of a modified system which leads to two Groups being enormous compared to the other six and this leads to poorly conceived figures. The TGCS is "shoe-horned" into the manuscript and does not add anything to the overall message. Maybe the paper would be better served by looking at purely patients demographics and characteristics and how they contributed to the decrease in the rate of CS. Use of the TGCS in Table 2 is particularly unnecessary. Using Group 1+2 simply cannot be compared to other groups in the system. The whole point of the TGCS is that the groups mutually exclusive and unrelated to each other. For example saying that a patients Odds Ratio of having a CS in Group 6 (Nulliparous Breech) vs Group 1+2 = 141 adds absolutely nothing the papers message. The inclusion of p-values in a sample size such as this is also not necessary as they are almost routinely <0.0001. We would have loved to have the ten groups in the Robson Classification, or even more and include the sub-groups, to perform the analysis. However, after all reasonable efforts, even crossing three different population-data registries, this was the best that we could obtain. 

As stated before, the scientific paper published by PLOS ONE “A Systematic Review of the Robson Classification for Caesarean Section: What Works, Doesn’t Work and How to Improve It”, states that the ten groups constitute the backbone of the Robson classification; however, they also find that many studies subdivide or merge Robson groups. In our case, the CMBD national registry is the most powerful registry of individual data at a national level in health science. We believe that utilising the CMBD for the analysis of CS according to Robson Classification is an opportunity of great calibre, which should not be eradicated for not having the 10 groups. 

We strongly believe that the TGCS is the cornerstone of our analysis and, for all the reasons mentioned in the comment above, we would like to keep it. 

In regards to Table 2, we understand that comparing Groups 1 and 2 versus Group 6 might not make much sense, but we consider it is useful and worth including, to see the behaviour of each group once the analysis is adjusted for all of the other variables. 

Nevertheless, if the Editor finds that Table 2 would be improved if Robson Classification is removed, as it is in an Excel attachment it can easily be removed and we will accept that change. N/A. 

The figures are also rendered useless by the use of the modified TGCS. Figure 2 is particularly not worthwhile. Figure 2 shows the relative size of obstetric population in %. 

The Robson Classification Implementation Manual (WHO) suggests how data should be reported. We followed those recommendations and presented data accordingly: group size, CS rate, relative size of obstetric population, and absolute contribution. 

We consider that by using visual aids we can contribute to a better and easier understanding from the reader, rather than using the classical results table. Specifically, this was because we wanted to evidence the tendency in each group; to do so, we decided to translate it into graphics. No changes made. 

Reviewer #4 Author response Changes made

Population: it is not fully clear in the methods the issue of missing data. In page 4 line 80, authors state that 9.1% of the women had missing data. It is, nonetheless, not clear to me the inclusion/exclusion criteria. What were the variables that were considered essential for inclusion of the women in the analysis? The variables necessary for the Robson classification? All variables used in the analysis? Please clarify in the Methods section. Authors could considered a flowchart showing the process of the exclusion of women to arrive to the final sample size of their analysis (this could be shown in the first paragraph of the results). Thank you for the comment; the sentence might be misleading. It has helped us to look back at the numbers and rectify. The main variable that allowed us to identify women in the registry was “delivery”. As for clarification, we stated that pregnancies < 22 weeks were excluded from the study. This statement, however, was erroneous. The CMBD only uses the variable “delivery” for deliveries after 22 weeks. We have now removed this statement.

The variable that was considered essential for the study was: caesarean section yes/no. We had 9.1% of omissions due to this variable, which reduced the sample to 210,241.

We have now included a flowchart. 

 Page 4, line 79. 

From the total 330,851 (100%) deliveries occurred after 22 weeks’ gestation, only those that occurred at the 44 publicly funded hospitals offering maternity care in Catalonia were included. As a result, 231,020 (69.83%) deliveries met the eligibility criteria for this research. However, information regarding the type of delivery, whether it was vaginal or caesarean section, was considered a sine qua non variable for the study. Thus, missing data in 9.1% of the women regarding outcome (vaginal/CS), reduced the potential sample size to 210,241.

Population: Consider adding the proportion of births occurring in the public health system among all births in Catalonia as an indication for the reader of the coverage of the analysis. Thank you very much, it has now been added. Please note the changes above. 

Data sources: one of the strengths of the analysis and manuscript is the richness of the data due to the availability and inclusion of multiple data sources in the analysis. Were there any inconsistencies found between the data in the three databases? If so, how were they resolved? For example, if I understand correctly, “parity” seem to be available through the CMBD and the E-CAP. Was there any hierarchy from where to select the data? Thank you for your comment. It has been tedious work to integrate the three data sources and we find your question very interesting. 

First of all, we would like to clarify that we did not find any inconsistency or discrepancy between the databases because we did not crosscheck the information. This was intention, for the following reason:

For Robson Classification, we are interested in the characteristics at the time the woman is admitted for delivery. E-CAP is the computer system that records information in the primary healthcare setting. By contrast, CMBD is the system that records the woman’s information from the hospital discharge documents, hence around delivery time. Therefore, since we wanted the information regarding the woman at the time of admission, looking at E-CAP would have been incorrect, since the latest documentation would have corresponded to the last midwifery appointment. 

That is why CMBD was our main data source, and all of the variables, apart from three – socioeconomic status, parity, and BMI – were obtained from there. 

Socioeconomic status: CMBD does not collect information regarding socioeconomic status, so we had to access RCA in order to obtain this information. 

Parity: It is not available through the CMBD, so in order to obtain this information we had to access E-CAP.

BMI: This is not a well reported variable in the CMBD: only 101 women out of 231,020 had linkage to the dx. overweight (BMI 25-29.9). However, the reason for this is because women admitted to the labour ward are not weighed on admission. This is not a recommended practice because there is no clinical reason to do so, and so it is not recommended in the clinical guidelines. For that reason, we accessed the E-CAP database to obtain BMI information from the booking appointment; this is common practice and is recommended in the protocol. No changes made. 

Variables: page 6 line 117-122: Some complications that may be important determinants of mode of birth are not in the list. Eclampsia is listed in Table S1 but I wonder about others such as infections of any type, haemorrhage or stillbirths. Was there no data on these complications available? Thank you for the question. Considering which conditions should be added to the “pregnancy complication” list was a meticulous and complex task. Since there is no specific or agreed local or international classification of high-risk pregnancies, we agreed upon this list based upon our own experiences and on advice from clinicians. 

We wanted to ensure that none of the complications included in the list could have been a direct cause of the management of the labour. We looked at the general CIE-9 diagnosis and each of the sub-groups. We only considered those that specifically belonged to the antepartum period. We also looked at how many people had a diagnosis we agreed on the list (this gave us an idea of how clinicians tended to report the complications). For infections, we included viral diseases in general, plus HIV specifically. Haemorrhage was included under the diagnosis of placenta praevia: haemorrhage from placenta praevia, premature separation of placenta, antepartum haemorrhage associated with coagulation defects (in the Catalan CMBD the description of the diagnosis allowed us to identify it was antepartum). Regarding stillbirths, we had many discussions. We could not identify whether this was prior to or during labour, so we could not include as such. 

Nevertheless, there is another, miscellaneous diagnosis, labelled “Other current conditions complicating pregnancy”, which we realised was widely used. Many conditions are recorded here: lupus, asthma, cancer, pneumonia, amongst others. We then determined that it would be useful to add this variable, because it would help us to include all those pathologies that we did not include through our list. No changes made.

Variables: the variable “socioeconomic status” seems to be “income”, correct? If so, consider calling it “income”. Also, I suggest that authors clarify/define the categories. Particularly, it is not clear what “disadvantaged populations” are and what marks the difference between the group with income <18,000 and the group under disadvantage populations (assuming they are mutually exclusive). Thank you for your comment, as mentioned in another comment from another reviewer, we have now included a clarification on how the socioeconomic status variable is constructed. 

Regarding the term “disadvantaged population”, we did have many discussions in our group in regards to this. We did some research before we finally agreed on this term. We found it is a frequently used term in academic/scientific papers, which is why we also adopted it. Page 6, line 122. The variable socioeconomic status is a set variable within the database that arises through pharmaceutical co-pay estimations. There are four co-payment groups: 1) those exempt from co-payment, (disadvantaged population; individuals receiving some form of universal pension scheme, those who no longer receive unemployment allowance, or those who no longer receive unemployment benefit and do not qualify for unemployment allowance); 2) those with annual earnings of less than 18,000€; 3) those with annual earnings between 18,000€ and 100,000€; and 4) those with annual earnings over 100,000€. It is predetermined by the system and does not allow further disaggregation.

Results, Page 7 line 156 and 157: authors describe decreases and increases as “reasonable”. Please clarify what “reasonable” is or delete. I think it will be appropriate to mention in the results those changes that are statistically significant. Thank you for your comment, we entirely agree. Page 8, line 176: In addition, the decrease in the proportion of nulliparous women and the increase in multiparous women with previous CS were statistically significant.

Results, page 9 line 168: “… particularly for those over thirty-five, who presented a 37% higher change….” According to Table 2, the 37% higher risk is for women 35-39.9 years of age not for all those above 35 years. Thank you very much for pointing out this error. It has now been corrected. Page 9, line 187. Adjusting for sociodemographic, institutional, and obstetric factors, the age differences remain similar. In particular, women between 35 and 40 presented a 37% higher chance of CS, and women over 40, twice the likelihood of having a CS than women aged between 25 and 30 (Table 2).

Results, page 11 line 194-196: “Within the population sample, the majority of women carried single babies at term and belonged to 195 groups 1+2 and 3+4 (31.45% and 48.88%, respectively)….”. It should be added “in cephalic presentation”. Correct, thank you for the contribution. Page 11, line 215. Within the population sample, the majority of women carried single babies at term in cephalic presentation and belonged to groups 1+2 and 3+4

Discussion, page 15 line 256-257: authors acknowledged under the limitations that cephalic presentation was diagnosed in the absence of a direct registration of a diagnoses of breech or transverse lie. I think this information should come also under the Methods section with all other definitions or methodological particularities of the variables. We entirely agree, we have now included this in the material and methods section, under the definition of variables. 

 Page 5, line 105. The variable foetal presentation was not recorded within the databases, so it had to be created ad hoc, thus considering cephalic presentation in the absence of breech or transverse lie.

Discussion, page 17 line 289-292: authors discussed their CS rate in Robson Group 5 which is 54.7%. On the basis of available evidence, this is a relatively low CS rate for this group. I wonder if authors could discussed if there is any specific policy for TOLAC in Catalonia and if they think this CS rate in Group 5 is homogeneous across the region or, on the contrary, is likely to suffer great variations between settings. Although it is not validated, the Robson Classification Implementation Manual published by WHO (https://www.who.int/reproductivehealth/publications/maternal_perinatal_health/robson-classification/en/) initially suggests that “Rates of 50-60% in Group 5 are considered appropriate provided you have good maternal and perinatal outcome”. Yes, thank you for pointing this out. We did realise this, but unfortunately, we do not know the reason for it. Currently there are no specific strategies or policies for TOLAC, apart from offering TOLAC to all suitable women. 

Overall, regarding distribution of CS in Group 5, we could summarise the following:

CS rate <50%: 14 hospitals.

CS rate between 50–60%: 17 hospitals.

CS rate >60%: 13 hospitals. 

For Group 5, if we look at the CS % according to hospital level:

Level I: 54.47%

Level IIA: 53.71%

Level IIB: 50.93%

Level IIIA: 59.52%

Level IIIB: 54.21%

We can see there are no major differences according to the level. 

We can say there are also no major discrepancies across the region. It should be noted that those hospitals with the highest rates, also have the highest overall CS rates.

Within the group, we have thought about different hypotheses, but these are based on assumptions, and further evidence would be needed. One of the hypotheses arose when we looked at the parity of our population. The % of multipara women in the public system is higher than average, and the % of women with previous CS continues to rise although there has been a downwards tendency in the % of CS within the last few years. One possible hypothesis is that some women that may have attended the private sector for their first delivery may have ended up with a CS as rates are higher in private hospitals, and then moved to a public hospital for the delivery of their second child. If these women are moving to the public sector because they are aiming for a vaginal delivery, they will be eager to attempt TOLAC. 

In addition, another hypothesis we considered is that there has possibly been a lot of effort to provide TOLAC options for women, starting from the hospital boards, obstetric leaders, and the Department of Health. 

International guidelines have also promoted offering this choice to women and have evidenced the safety of trying for a vaginal birth. No changes made. 

Discussion, page 17 lines 293-298: authors discuss the CS rates in breech presentations acknowledging that the use of CS in breeches has been driven by the results of the Term Breech Trial. Important to note that the authors of the 2-year trial follow-up noted that ‘‘planned caesarean delivery is not associated with a reduction in risk of death or neurodevelopmental delay in children at 2 years of age’’ [Reference: Whyte H, Hannah ME, Saigal S, et al. Outcomes of children at 2 years after planned cesarean birth versus planned vaginal birth for breech presentation at term: the International Randomized Term Breech Trial. Am J Obstet Gynecol 2004; 191: 864–71.] Thank you for raising this interesting point. The study that we referred to concluded: “planned caesarean section is better than planned vaginal birth for the term foetus in the breech presentation; serious maternal complications are similar between the groups”. Although the study was later found to be “faulty” and the follow-up study two years later suggested “no associated risk of death or neurodevelopmental delay in children at 2 years of age”, the damage was already done; the norm of performing CS was already established. 

This is unfortunate, but we believe it highlights how “easy” is to medicalise childbirth, and how difficult becomes to “de-medicalise”. In addition, it should be noted that the “art of assisting vaginal breech births” was almost completely eliminated from medical and midwifery schools and from delivery rooms, which aggravated these consequences, as newly qualified professionals were not receiving training or hands-on experience in breech vaginal deliveries. 

As mentioned in the manuscript, it is only recently that some hospitals in Catalonia have resumed vaginal breech deliveries. No changes made. 

Discussion, page 17 line 299-303; authors discuss their hypothesis and the potential reasons behind the decrease of CS in Catalonia. In contemporary societies where fear of litigation and fear of damage reputations are significant factors underpinning the increase of CS, a 1.5 percent points of significant decrease is impressive, particularly at population level and in Robson Groups 1 through 4. I wonder if the authors would like to add in the discussion about this point or if they know of any on-going efforts in Catalonia to mitigate the fear among healthcare professionals. This would be very useful for other countries under similar circumstances. Thank you for your comment, we find it very interesting, but unfortunately, we do not have relevant information. 

The reduction of CS rate has been an internal quality indicator for many hospitals. The Department of Health has also been adamant about wanting to reduce this rate. Many hospitals have promoted education programmes in cardiotocography (CTG) which could also have had an influence. 

However, we do not know whether fear of litigation and fear of damaged reputation has played a role. 

We also consider the possibility that fear of litigation can go one of two ways. Scenario one, where a woman having a vaginal traumatic delivery, with consequences to the mother or the baby, claims they were not offered a CS. Or alternatively, scenario two, where a woman who wanted a vaginal delivery ends up with a CS, but believes could have had a vaginal delivery if her care had been different. No changes made. 

Discussion: Of interest is the reverting pattern shown on CS rates in Catalonia with regard to income level. The normal pattern seen in the literature tends to show higher CS rates in the wealthier women whereas this study shows the contrary. I understand that most of the wealthiest women would be giving birth in the private sector and thus, if incorporated in this analysis, they could have increased the CS rate of the wealthier groups in this paper. Do authors think that this is the only explanation for this pattern in Catalonia or they think they may be something else underlying the pattern? Thank you for your comment, we are more than happy to share our views, since we did have deep discussions around these results. 

Our hypothesis is that some/many wealthy women that attend public hospitals, would prefer to go to a public hospital (despite even having private health insurance). We think these might be women with a particular profile; highly educated, with access to information, who are worried about higher CS rates in private hospitals and attend public hospitals because they know the rates of vaginal birth are higher, etc. In addition, we think that even though antenatal classes are already included in the public system for free, these might be women who have also attended private classes for hypnobirthing, yoga or pilates, for instance, and are in some way more” prepared” for and with higher willingness to have a natural childbirth. N/A.

---

## [Decision Letter · Decision Letter 1]

2 Jun 2020

Caesarean section trends in Catalonia between 2013 and 2017 based on the Robson classification system: a cross-sectional study.

PONE-D-20-00730R1

Dear Dr. García-Altés,

We are pleased to inform you that your manuscript has been judged scientifically suitable for publication and will be formally accepted for publication once it complies with all outstanding technical requirements.

With kind regards,

Calistus Wilunda, DrPH

Academic Editor

PLOS ONE

Additional Editor Comments (optional):

Reviewers' comments:

Reviewer's Responses to Questions

**Comments to the Author**

1. If the authors have adequately addressed your comments raised in a previous round of review and you feel that this manuscript is now acceptable for publication, you may indicate that here to bypass the “Comments to the Author” section, enter your conflict of interest statement in the “Confidential to Editor” section, and submit your "Accept" recommendation.

Reviewer #1: All comments have been addressed

Reviewer #2: All comments have been addressed

Reviewer #4: All comments have been addressed

2. Is the manuscript technically sound, and do the data support the conclusions?

Reviewer #1: Yes

Reviewer #2: Yes

Reviewer #4: Yes

3. Has the statistical analysis been performed appropriately and rigorously? 

Reviewer #1: Yes

Reviewer #2: Yes

Reviewer #4: I Don't Know

4. Have the authors made all data underlying the findings in their manuscript fully available?

Reviewer #1: Yes

Reviewer #2: Yes

Reviewer #4: Yes

5. Is the manuscript presented in an intelligible fashion and written in standard English?

Reviewer #1: Yes

Reviewer #2: Yes

Reviewer #4: Yes

6. Review Comments to the Author

Reviewer #1: My comments have been addressed and have answered any queries that I have. I thank the authors for their transparency and recommend publication.

Reviewer #2: I have one observation with BMI. clinical experience or any assumption should not put while selecting variable. other than pre-pregnancy BMI, none of the maternal BMI is a good predictor of CS outcome. Since you do not include BMI in final model, I do not want to hang on this issue. However, it is worth to mention BMI taken during 1st ANC visit was included as co-variates considering 1st ANC taken before 12 weeks. It is also worth to mention either the BMI was subjective or from direct anthropocentric measurement

Reviewer #4: The authors have responded and clarified my questions and comments. I have no further comments to the authors.

7. PLOS authors have the option to publish the peer review history of their article (what does this mean?). If published, this will include your full peer review and any attached files.

Reviewer #1: No

Reviewer #2: No

Reviewer #4: No

---

## [Editor Report · Acceptance letter]

5 Jun 2020

PONE-D-20-00730R1 

Caesarean section trends in Catalonia between 2013 and 2017 based on the Robson classification system: a cross-sectional study. 

Dear Dr. García-Altés:

I'm pleased to inform you that your manuscript has been deemed suitable for publication in PLOS ONE. Congratulations! Your manuscript is now with our production department. 

Kind regards, 

on behalf of

Dr. Calistus Wilunda 

Academic Editor

PLOS ONE